



# Characterization of organic nitrate constituents of secondary organic aerosol (SOA) from nitrate-radical-initiated oxidation of limonene using High-Resolution Chemical Ionization Mass Spectrometry

Cameron Faxon[1], Julia Hammes[1], Ravi Kant Pathak[1], Mattias Hallquist[1]

[1]Department of Chemistry and Molecular biology, University of Gothenburg, Göteborg, SE-41258, Sweden

**Abstract**: The gas phase nitrate radical ($NO_3^\bullet$) initiated oxidation of limonene can produce organic nitrate species with varying physical properties. Low-volatility products can contribute to secondary organic aerosol (SOA) formation and organic nitrates may serve as a $NO_x$ reservoir, which could be especially important in regions with high biogenic emissions. This work presents the measurement results from flow reactor studies on the reaction of $NO_3^\bullet$ and limonene using a High-Resolution Time-of-Flight Chemical Ionization Mass Spectrometer (HR-ToF-CIMS) combined with a Filter Inlet for Gases and AEROsols (FIGAERO). Major condensed-phase species were identified, and the identity and volatility properties of the most prevalent organic nitrates in the produced SOA were determined. Analysis of multiple experiments resulted in the identification of several dominant species (including $C_{10}H_{15}NO_6$, $C_{10}H_{17}NO_6$, $C_8H_{11}NO_6$, $C_{10}H_{17}NO_7$, and $C_9H_{13}NO_7$) that occurred in the SOA under all conditions considered. The observed and expected (listed) products (associated with the Master Chemical Mechanism (MCM) limonene mechanism) were compared, and many non-listed species were identified. Additionally, the formation of dimers was consistently observed and these species resided almost completely in the particle phase. The identities of these species are discussed, and formation mechanisms are proposed. Cluster analysis of the desorption temperatures corresponding to the analyzed particle-phase species yielded at least five distinct groupings based on a combination of molecular weight and desorption profile. Overall, the results indicate that the oxidation of limonene by $NO_3^\bullet$ produces a complex mixture of highly oxygenated monomer and dimer products that contribute to SOA formation.

## 1 Introduction

Oxidation of gas-phase organic species contribute significantly to particle formation and growth (Hallquist et al., 2009; Smith et al., 2008; Wehner et al., 2005), and thus a thorough understanding of secondary organic aerosol (SOA) formation mechanisms is important for the accurate estimation of its impact on the climate system (Kanakidou et al., 2005).

Secondary organic aerosols form primarily via the photooxidation of volatile organic compounds (VOCs), yielding less volatile products, which can then partition into the condensed phase (Hallquist et al., 2009; Kroll and Seinfeld, 2008), especially when pre-existing aerosols (e.g., inorganic seed particles) are present (Kroll et al., 2007). The products resulting from atmospheric oxidation may be classified as low volatility, semi-volatile, and intermediate volatility OCs, i.e.,



LVOCs, SVOCs, and IVOCs, respectively (Donahue et al., 2012; Jimenez et al., 2006; Murphy et
al., 2014). In addition, extremely low volatility OCs (i.e., ELVOCs) contribute significantly to
aerosol formation and early growth (Ehn et al., 2014; Jokinen et al., 2015). The photo-oxidation
of VOCs by the primary atmospheric oxidants, $O_3$ and $^{\bullet}OH$, has been extensively investigated
(Cao and Jang, 2008; Hallquist et al., 2009; Kanakidou et al., 2005; Kroll and Seinfeld, 2008).
Although less studied than the photo-oxidation of VOCs, the reaction of VOCs with the nitrate
radical ($NO_3^{\bullet}$) and the resulting formation of organic nitrates are also important, especially for
nocturnal chemistry (Brown and Stutz, 2012; Perring et al., 2013; Roberts, 1990). Significant
concentrations of these nitrates have been detected in the gas and condensed phases in both field
and laboratory studies (Ayres et al., 2015; Beaver et al., 2012; Bruns et al., 2010; Lee et al., 2016;
Paulot et al., 2009; Rindelaub et al., 2014, 2015, Rollins et al., 2012, 2013).
Organic nitrates ($RONO_2$) and organic peroxy nitrates ($RO_2NO_2$), such as peroxy acetyl
nitrate (PAN), may also form in the atmosphere (Roberts, 1990; Singh and Hanst, 1981; Temple
and Taylor, 1983). $RO_2NO_2$ may form via the reaction of organic peroxy nitrates ($RO_2^{\bullet}$) with
$NO_2$, while $RONO_2$ may form directly through either the reaction of $RO_2^{\bullet}$ with NO or the
reaction of unsaturated VOCs with $NO_3^{\bullet}$. Reactions 1–2 show the typical formation pathway of
organic nitrates and peroxy nitrates formed from the reactions of $NO_x$ with $RO_2^{\bullet}$ originating from
a generic VOC, R
$$RO_2^{\bullet} + NO \xrightarrow{\text{(a)}} NO_2 + RO^{\bullet} \qquad\qquad \text{(R1a)}$$
$$\xrightarrow{\text{(b)}} RONO_2 \qquad\qquad\qquad \text{(R1b)}$$
$$RO_2^{\bullet} + NO_2 \longrightarrow RO_2NO_2 \qquad\qquad \text{(R2)}$$

Secondary organic aerosol-precursor VOCs arise mainly from the emission and reaction of
biogenic VOCs (BVOCs) (M. Hallquist et al., 2009), with up to 90% of the global VOC budget
originating from biogenic sources (Glasius and Goldstein, 2016; Guenther et al., 1995). Isoprene,
the main constituent of global BVOC terrestrial emissions (600 Tg $yr^{-1}$) (Guenther et al., 2006), is
highly reactive with $^{\bullet}OH$, $O_3$, and $NO_3^{\bullet}$ (Atkinson et al., 1995; Hallquist et al., 2009). However,
monoterpenes typically have higher SOA yields than isoprene (Carlton et al., 2009; Presto et al.,
2005b) and regarding atmospheric emissions, α-pinene, β-pinene, and limonene constitute the
main monoterpenes emitted into the atmosphere (Guenther et al., 2012). In addition to its high
emission rates, limonene is especially interesting as a model BVOC, due to its relatively high
reaction rates (Ziemann and Atkinson, 2012) and occurrence in indoor environments, owing to
emission sources, such as air fresheners and other household products (Wainman et al., 2000).
The reactions and mechanisms of α-pinene and β-pinene oxidation have been more
thoroughly studied (Bonn and Moorgat, 2002; Fry et al., 2009; Perraud et al., 2010; Presto et al.,
2005a, 2005b) than those associated with limonene. Several studies have focused on the
ozonolysis of and SOA formation from limonene (Baptista et al., 2011; Jiang et al., 2013;
Jonsson et al., 2006, 2008a; Leungsakul et al., 2005; Pathak et al., 2012; Sun et al., 2011;





Youssefi and Waring, 2014; Zhang et al., 2006). $NO_3^{\bullet}$ oxidation of limonene and the resulting
organic nitrates that may contribute to SOA formation have, however, rarely been investigated
(Fry et al., 2011, 2014; Hallquist et al., 1999; Spittler et al., 2006). In relation to the reaction with
$NO_3^{\bullet}$, major non-nitrate products of limonene (including endolim) have been identified, but
significant SOA formation was preceded by the occurrence of multiple unidentified nitrates
(Hallquist et al., 1999; Spittler et al., 2006). Moreover, although mechanistic models and
molecular identities of these products have been proposed, direct measurement and identification
thereof have yet to be reported. Further elucidation of the mechanisms governing and products
generated by the reactions of limonene and $NO_3^{\bullet}$ are warranted, since organic nitrates from
BVOCs (including limonene) have been consistently observed in field studies (Ayres et al., 2015;
Beaver et al., 2012; Lee et al., 2016, 2014b; Perring et al., 2009).

Additionally, the contribution of low-volatility products to the SOA mass may increase
with the formation of dimers from aerosol components generated by VOC oxidation. Numerous
dimers or oligomers have been found in SOA generated by monoterpene species (e.g.
Emanuelsson et al., 2013; Kourtchev et al., 2014, 2016; Kristensen et al., 2016; Müller et al.,
2007; Tolocka et al., 2004). However, the speciation of observed dimers and oligomers from
organic nitrates, especially with respect to detailed formation mechanisms, has rarely been
reported.

Here we report the chemical composition of low-volatility gas and aerosol-phase species,
formed from mixtures of $N_2O_5$ and limonene, as measured by a High Resolution Time-of-Flight
Chemical Ionization Mass Spectrometer (HR-ToF-CIMS) coupled to a Filter Inlet for Gases and
AEROsols (FIGAERO) inlet (Lopez-Hilfiker et al., 2014). The objectives of this work were
three-fold namely, to: (i) determine the molecular formula of major nitrate species, produced
from the reaction of limonene with $NO_3^{\bullet}$, that could contribute significantly to SOA formation
and growth, (ii) compare the distribution of measured products to that of the expected products
(formed via the Master Chemical Mechanism (MCM)) to identify any discrepancies in the
mechanistic understanding of nitrate formation from limonene, and (iii) categorize, via cluster
analysis, the thermodynamic desorption data measured for selected condensed-phase species.
**2 Methods**
**2.1 Experimental setup**
Experiments were performed in the Gothenburg Flow Reactor for Oxidation Studies at
low Temperatures (GFROST) at the University of Gothenburg. In previous studies, this facility
was used for studying the impact of relative humidity, OH-scavengers, and temperature on SOA
formation via monoterpene ozonolysis (Emanuelsson et al., 2013; Jonsson et al., 2008a, 2008b),
its volatility properties (Pathak et al., 2012), and dimer formation during the ozonolysis of α-
pinene (Kristensen et al., 2016). The inflow of zero air and the reagents is fixed at a total flow of
1.6 L per min (LPM). The experiments are all run at low RH (≤1%) and a constant temperature of
20°C. To catch only the center portion of the laminar flow and avoid unnecessary interference





from wall effects, samples are taken through a cone at the end of the reactor at 0.95 LPM. The
average residence time of the sampled portion of the mixture is 240 s. Due to the flow
restrictions, a make-up flow of zero air is added to the sample, immediately after the outlet, prior
to being sampled by the instruments. The amount of dilution flow necessary is constrained by the
flow required by the HR-ToF-CIMS. Figure 1 shows a diagram of the experimental setup.

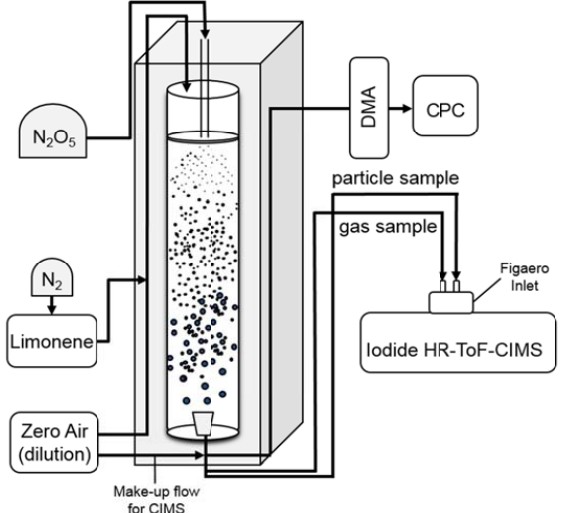

**Figure 1.** Diagram of experimental setup of GFROST during experiments.
Gas and particle-phase products were measured using a High-Resolution Time-of-Flight
Chemical Ionization Mass Spectrometer (HR-ToF-CIMS) coupled to a Filter Inlet for Gases and
AEROsols (FIGAERO) (Lopez-Hilfiker et al., 2014). The HR-ToF-CIMS can be operated in
either negative- or positive-ionization modes, using various reagent-ion sources. CIMS
measurement techniques have previously been employed for the measurement of organic nitrate
products of monoterpenes (Beaver et al., 2012; Paulot et al., 2009) using multiple reagent ions
(Lee et al., 2014a). In this work, the HR-ToF-CIMS was operated using negative Iodide ($I^-$) ion
as the reagent in all experiments. Dry UHP $N_2$ was passed over a permeation tube containing
liquid $CH_3I$ (Alfa Aesar, 99%), and $I(H_2O)_n^-$ ions were generated by directing the flow over a
$^{210}$Po radioactive source. Reaction products (e.g., species X) were identified by their
corresponding cluster ions, $XI^-$, thereby allowing the collection of whole-molecule data. The
reagent and sample flowed into the Ion-Molecule Reaction (IMR) chamber of the instrument at a
nominal individual rate of 2 LPM. The IMR was temperature-controlled at 40°C and operated at
a nominal pressure of 200 mbar. With $I^-$ ionization, the sensitivity of a detected species (i.e., hz
$ppt^{-1}$) can vary significantly with relative humidity (Lee et al., 2014a). However, the experiments
were all performed at low RH (≤1%) and, hence, the same sensitivity was realized for all the
conditions considered.



The FIGAERO inlet was used during the experiments, and particles were collected on a
Zefluor® PTFE membrane filter. The aerosol sample line and gas sample line were composed of
12 mm copper tubing and 12 mm Teflon tubing, respectively. The inlet was operated in regular
cycles – 1 h of gas-phase sampling and simultaneous particle collection, followed by a 1-h period
where the filter was shifted into position over the IMR inlet and the collected SOA was desorbed.
Desorption was facilitated by a 2 LPM flow of heated UHP $N_2$ over the filter. The temperature of
the $N_2$ was increased from 20 to 200°C in 50 min (3.5°C min$^{-1}$), and a subsequent 10-minute
temperature soak was performed to ensure complete removal of the remaining organic material
that volatilizes at 200°C. The measured species were distinguished based on their thermal
properties via the resulting desorption time-series profiles, hereafter referred to as thermograms.
Temperature gradients of >3.5°C min$^{-1}$ have been used in previous studies, but, in this work, a
lower gradient was used to enable optimum thermal separation (Lee et al., 2014a; Lopez-Hilfiker
et al., 2014). The HR-ToF-CIMS was configured to measure singly charged ions with a mass-to-
charge ratio (*m/z* or Th) of 7–720. Particles were contemporaneously sampled directly at the
outlet of the flow reactor, through a ¼″ stainless steel 1 m sample line, by a Scanning Mobility
Particle Sizer (SMPS). The SMPS measured the number-size distribution used for estimating the
mass concentrations, based on the assumption of spherical particles with a density of 1.4
(Hallquist et al., 2009). In all cases, SOA was generated via nucleation and growth rather than by
using seed particles.

## 2.2 Reagent preparation

$N_2O_5$ was synthesized by reacting ≥20 ppm $O_3$ with pure $NO_2$ (98%, AGA Gas) in a glass
vessel and then passing the flow through a cold trap maintained at -78.5°C by dry ice. The
resulting white solid showed signs of yellowing, due to nitric or nitrous acid contamination, only
when exposed to moisture (e.g., ambient lab air). The solid $N_2O_5$ was transferred to a diffusion
vial fitted with a capillary tube (inner diameter: 2 mm). The $N_2O_5$ diffusion source was held at a
constant temperature (-23 °C), and the gravimetrically determined mass loss rate remained steady
($r^2$ value: 0.97–0.98) for several weeks. A similarly characterized d-limonene (Alfa Aesar, 97%)
diffusion source was held at temperatures ranging from 8.5 to 31.5°C and, using Gas
Chromatography–Mass Spectrometry (GC-MS; Finnigan/Tremetrics), diluted flow-reactor
concentrations (15, 45, 92, and 150 ppb)..
Experiments were performed over a range (1.0–113) of $N_2O_5$/limonene ratios (see Table 1
for a summary of experimental conditions). For each set of conditions in the flow reactor,
sampling was performed over a period of 6–12 h to ensure stability of conditions (e.g., gas-phase
signals, total SOA mass) and repeatability of the FIGAERO thermal-desorption cycles.



**Table 1.** Experimental conditions considered in this study.

| # | $N_2O_5$ (ppb) | Limonene (ppb) | $N_2O_5$ / Limonene | Average SOA Mass ($\mu g\ m^3$) |
|---|---|---|---|---|
| 1 | 160 | 15 | 10.7 | 8.1 |
| 2 | 95 | 40 | 2.4 | 8 |
| 3 | 95 | 15 | 6.3 | 12 |
| 4 | 95 | 15 | 6.3 | 8 |
| 5 | 95 | 40 | 2.4 | 10 |
| 6 | 95 | 95 | 1.0 | 12 |
| 7 | 1700 | 15 | 113.3 | 7 |
| 8 | 1700 | 40 | 42.5 | 11 |
| 9 | 1700 | 95 | 17.9 | 43 |
| 10 | 1700 | 150 | 11.3 | 95 |
| 11 | 850 | 95 | 8.9 | 25 |
| 12 | 850 | 150 | 5.7 | 47 |


**2.3 CIMS data-analysis methods**
Data obtained from the HR-ToF-CIMS was analyzed using the Tofware
(Tofwerk/Aerodyne) analysis software written in Igor Pro (WaveMetrics). High-resolution
analysis allowed for ion identification with a resolution of ∼4000 (m/Δm). Identified species were
cross-checked with predicted species generated via the MCM v3.3.1 limonene mechanism
(Saunders et al., 2003) and the corresponding theoretical product distribution was compared with
the measured distribution. For several ions, product formulas in the MCM were used as the major
parameter for ion identification at a given *m/z*. However, this identification scheme resulted in the
misidentification of several ions. The identification of high-mass ions (*m/z* > 500) was
complicated by the fact that the number of possible formulas increases rapidly with increasing
mass and carbon number of the ions. Nevertheless, the high accuracy of fits (≤5 ppm), where the
identities of expected product ions were corroborated by the fits of expected isotopes, reduced
uncertainties stemming from the mass calibration and provided reliable ion identifications. To
further ensure the accuracy of the identities of high-mass ions, the fits of the identified ions were
compared over all experiments.
The high-resolution ion data was further analyzed with Python 3.5.2 using the pandas
(McKinney, 2010, 2011) and NumPy (Van Der Walt et al., 2011) packages, and peaks in the ion
thermograms were identified using an implementation of the PeakUtils package (v1.0.3,
http://pythonhosted.org/PeakUtils/). For each experiment, the temperature ($T_{max}$) corresponding
to the peak signal of each ion observed during the desorption of SOA particles was identified.
Furthermore, a secondary temperature ($T_{max,2}$) was identified when double-peak behavior was
observed.





**2.4 Cluster-analysis methods**

Cluster analysis, performed via the K-Means algorithm (scikit-learn machine learning
package; Pedregosa et al., 2011), was used to distinguish, based on their elemental composition
and thermodynamic behavior ($T_{max}$), groups of ions observed during SOA desorption. This
algorithm, utilizing a random seeding approach (Arthur and Vassilvitskii, 2007), was chosen due
to the superior cluster separation realized after comparing several algorithms, including affinity
propagation and mean-shift clustering. The solution of the K-Means algorithm is obtained
through the minimization of an inertia function (see Eq. 1) $\Phi$, which is equivalent to the sum of
the mean-squared distance between all samples and their corresponding cluster centroid, c
(Arthur and Vassilvitskii, 2007; Raschka, 2016). Here, $x^{(i)}$: sample (e.g., carbon number, oxygen
number, $T_{max}$) in a set of $n$ samples, $c^{(j)}$: cluster center of cluster $j$ in a set of $k$ clusters, and $w^{(i,j)}$:
weighting coefficient ($w^{(i,j)} = 1$ if $x^{(i)}$ is in cluster $j$, $w^{(i,j)}=0$ otherwise).
$$\phi = \sum_{i=1}^{n} \sum_{j=1}^{k} w^{(i,j)} \left\| x^{(i)} - c^{(j)} \right\|^2 \qquad\qquad (1)$$

The quality of the cluster separation was assessed through a silhouette score (Rousseeuw,
1987), which allows comparison of the intra-cluster and inter-cluster distances and, for a sample
$i$, is determined from:
$$s(i) = \frac{b(i) - a(i)}{max\{a(i), b(i)\}} \qquad\qquad (2)$$

where, a($i$): average distance, or dissimilarity, between point $i$ and each point within its
own cluster and b($i$): average dissimilarity between point $i$ and all points within the nearest
neighboring cluster. The value of s($i$) ranges from -1 to 1 and reflects the quality of the clustering
with respect to the separation between members of each cluster. For example, a score of ~1
indicates that the point is relatively far away from the nearest neighboring cluster, while a score
of 0 suggests that the cluster separation is roughly equivalent to that of cohesion clusters; that is,
$a(i) \approx b(i)$. For all points within a clustered dataset, an average silhouette score can indicate the
adequacy of the cluster separation for a given number of clusters.

Detected ions were clustered based on their molecular weight (MW), elemental numbers
($n_c$, $n_H$, $n_O$, $n_N$), and $T_{max}$ values. Compared with the other variables, MW and the carbon number
exhibited the highest correlation with $T_{max}$. Clustering the ions based on these three variables
yielded the best separation with respect to mass and $T_{max}$ of the ions. Input variables were scaled
to values between 0 and 1 (based on their respective range of input values) to prevent any bias
associated with the relative magnitude of each variable (e.g., MW >> $n_C$).





## 3 Results and discussion

### 3.1 Characterization of mass spectra from SOA and identification of species

Products in both the gas and condensed phases were identified by analyzing HR-ToF-CIMS data collected under various experimental conditions (Table 1). In each sampling regime, major products were readily identifiable, and only modest or negligible fragmentation occurred with application of the soft ionization technique. The mass-to-charge (*m/z* or Th) values of the most prominent ions of species detected in the collected aerosol were determined from the average mass spectra obtained during desorption cycles. The results revealed two distinct regions consisting of several clusters of elevated ion signals (Fig. 2). These regions corresponded to aerosol samples considered in all experiments (Table 1), and the relative signal intensities varied with the amount of limonene and $N_2O_5$ present in the system. The occurrence of ions in these regions indicates a prevalence of lower-mass monomer species and higher-mass dimer species. These results are analogous to those of previous ozonolysis studies, where highly oxygenated multifunctional (HOM) molecules from monoterpene oxidation were observed using a nitrate HR-ToF-CIMS (Ehn et al., 2014; Jokinen et al., 2015; Mentel et al., 2015). Figure 2 shows an average mass spectrum corresponding to four sequential 1-h desorption cycles of 12-μg m$^{-3}$ SOA samples from a reaction mixture with a $N_2O_5$ to limonene ratio of 2.4.

In total, 198 of the identified organic ions constituted significant fractions of the aerosol samples, but most of the signal emanated from only ~25% of these species. The dominant species were identified by averaging the desorption-time series of all experiments and extracting the top 75[th] percentile (by averaging the signal during desorption) of the monomer and dimer ions. The resulting set of ions consisted of 52 molecular species that accounted for 76% of the organic signal during desorption, while the top 90[th] percentile of ions (20 ions) accounted for 56%. This 52-ion set consisted of 28 monomers (C = 7–10) and 24 dimers or oligomers (C = 11–20). On average, the top 75[th] percentile of monomers and the top 75[th] percentile of dimers accounted for 83% of the total monomer signal and 70% of the total dimer signal, respectively. A full list of ions and the composition of the 40[th], 75[th], and 90[th] percentile subsets can be found in the Supplementary Information.

The first, i.e., lower-mass region, of the two mass-spectra regions (see Fig. 2) occurred at *m/z* values ranging from 350 to 450. Several ions in this region matched the predicted molecular formulas associated with the MCM limonene mechanism, and the largest signals occurred for species consisting of 8–10 carbon atoms. E.g. the dominant ions occurring at *m/z* 360, 372, 374, and 390 (during desorption) corresponded to the iodide-cluster ions $C_8H_{11}NO_7I^-$, $C_{10}H_{15}NO_6I^-$, $C_{10}H_{17}NO_6I^-$, and $C_{10}H_{17}NO_7I^-$ (Fig. 2a). These correspond to the MCM species C727PAN and C731PAN, C923PAN, NLIMALOH and LIMALNO3, NLIMALOOH, respectively.





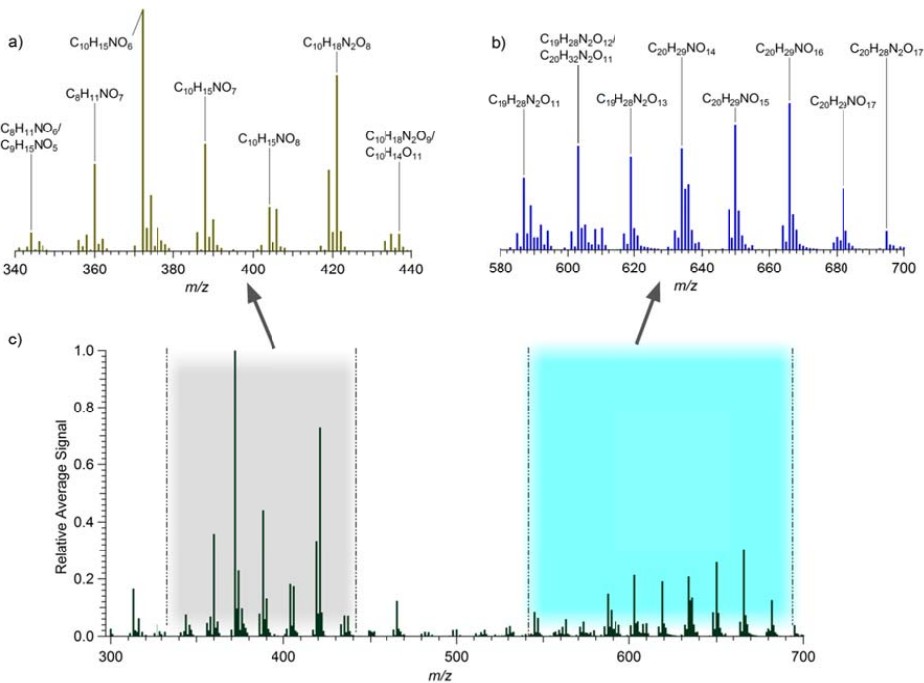

**Figure 2.** Representative average mass spectrum for the desorption of SOA collected during the experiments:
Identification of ions detected in the (a) monomer region ($m/z$ 340–440) and (b) dimer region ($m/z$ 580–700). (c)
Relative intensities and positions of the two regions detected in all aerosol samples. Data was obtained from four 1-h
desorption cycles of 12-µg m$^{-3}$ samples from a mixture with a $N_2O_5$/limonene ratio of 2.4. The un-clustered (i.e., not
clustered with I$^-$) $m/z$ of each ion is $-127$ $m/z$.
Elevated signals of monomer ions (e.g., $C_{10}H_{15}NO_7$ ($m/z$ 388), $C_{10}H_{15}NO_8$ ($m/z$ 404),
$C_{10}H_{17}NO_8$ ($m/z$ 406), and $C_{10}H_{15}NO_9$ ($m/z$ 420)), which are absent from the list of expected
products of the mechanism, also occurred in this region. These non-MCM species contributed, at
least modestly, to the total organic monomer signal, and MCM species accounted for only 43.5 ±
3.2% of the total monomer signal of all experiments. In contrast to the MCM species, the
monomers contain one nitrogen atom and more than six oxygen atoms.
Monomers with progressively more oxygenated species of the general formula
$C_{10}H_{15}NO_x$ were detected for x = 5–9 (i.e., $C_{10}H_{15}NO_5$–$C_{10}H_{15}NO_9$) with $C_{10}H_{15}NO_6$ being the
dominant species in both the aerosol and gas phase in most experiments. Ions with molecular
formulas containing two nitrogen atoms, for example, $C_{10}H_{16}N_2O_8$ ($m/z$ 419) and $C_{10}H_{18}N_2O_8$
($m/z$ 421), were also detected (Fig. 2a). Limonene and its primary products reacted only with
$NO_2$, $NO_3^•$, and $HNO_3$, yielding molecules that are most likely di-nitrate species, with additional
functional groups. Some of the measured species could either be peroxy nitrates or similar in
structure to peroxy acetyl nitrate (PAN). PAN-like species readily undergo thermal degradation



(Orlando et al., 1992; Tuazon et al., 1991; Wooldridge et al., 2010) at temperatures significantly
lower than those (100–200°C) used to heat the samples. Therefore, the number of intact PAN-like
ions reaching the detector could be significantly diminished relative to their prevalence in the
collected aerosol. However, the measured ions with a given formula may consist of peroxy-
nitrates, di-nitrates, and PAN-like species.
Similar to the highly oxygenated multi-functional species (HOMs) resulting from the
ozonolysis of monoterpenes (Ehn et al., 2014; Jokinen et al., 2015), including limonene, many of
the observed species could be classified as extremely low-volatility organic compounds (i.e.,
ELVOC, with the general formula $C_{10}H_{14-16}O_{7-11}$), which play a key role in SOA formation
(Donahue et al., 2012). Observations performed under ambient conditions during the 2013
Southern Oxidant and Aerosol Study (SOAS) revealed the presence of highly functionalized
particulate organic nitrates containing 6–8 oxygen atoms (Lee et al., 2016). In that work, these
species constituted 3% and 8% of sub-μm aerosol mass during daytime and nighttime hours,
respectively, and exhibited a distinct diurnal pattern, typically reaching peak concentrations
between midnight and the early-morning hours. The gaseous parent compounds were identified
as monoterpenes, using α-pinene. The similarity with ions from the $NO_3^{\cdot}$-initiated limonene
oxidation further emphasizes the importance of monoterpenes as precursors of organic nitrates.
Furthermore, the occurrence of such compounds in the ambient environment reinforces the notion
that the formation of the highly oxygenated limonene-derived organic nitrate species detected,
here, is important to the troposphere.
Through the MCM mechanism, elevated ion signals above $m/z$ 390 occurred without
formation of the corresponding products. As shown in Fig. 2b, the largest ion signals
corresponded to compounds with 19 and 20 carbons in the dimer region. $C_{20}H_{22}N_2O_8$ and
$C_{20}H_{29}NO_{17}$, which occurred at significantly elevated levels in all aerosol samples, constituted the
lowest- and highest-mass dimers, respectively (see Fig. 2 for other examples of $C_{19}$ and $C_{20}$ dimer
species). Many of these can be considered ELVOC species based on their respective formulas
(e.g., $C_{19-20}H_{28-32}O_{10-18}$ for dimers) and their partitioning behavior (i.e., they occurred only in the
aerosol phase, at levels slightly above background in the gas samples). $C_{19}H_{28}N_2O_x$ and
$C_{20}H_{29}NO_x$ were the most dominant families of $C_{19}$ and $C_{20}$ dimers, respectively. Taken together,
10 individual dimers from these two families were identified in all experiments.
The contributions of the 11 most prevalent ion families to the total desorbed organic
signal are summarized in Table 2. Average contributions are calculated from the mean signals for
each family relative to the total mean organic signal generated during all experiments.



**Table 2.** Peak desorption temperature ($T_{max}$) and the average contribution (over all experiments) to the organic signal
during SOA desorption for the most commonly observed product families. The number of monomer species in each
family that desorbed at only high temperatures is noted in parentheses.

| Class | # | Family | # Observed in Family | (N/C)×10 | Average Contribution | $T_{max}$ Range (°C) |
|---|---|---|---|---|---|---|
| **Monomers** | m1 | C10H15NOx | 5 (1) | 1.0 | 23.0% | 74.7 − 152.4 |
| | m2 | C10H18N2Ox | 2 (0) | 2.0 | 8.8% | 66.3 − 69.8 |
| | m3 | C10H16N2Ox | 5 (1) | 2.0 | 6.7% | 51.9 − 154.0 |
| | m4 | C10H17NOx | 5 (2) | 1.0 | 5.3% | 58.7 − 159.2 |
| | m5 | C8H11NOx | 3 (0) | 1.3 | 4.7% | 67.5 − 80.5 |
| | m6 | C9H13NOx | 4 (0) | 1.1 | 3.0% | 70.4 − 74.5 |
| | m7 | C9H15NOx | 4 (0) | 1.1 | 2.0% | 63.9 − 75.7 |
| **Dimers** | d1 | C20H29NOx | 4 | 0.5 | 7.1% | 100.2 − 154.0 |
| | d2 | C19H28N2Ox | 6 | 1.1 | 5.0% | 101.1 − 156.9 |
| | d3 | C20H27NOx | 4 | 0.5 | 2.8% | 100.5 − 151.3 |
| | d4 | C20H24N2Ox | 3 | 1.0 | 2.0% | 124.5 − 156.9 |


### 3.2 Characterization of identified ions via thermal properties

The desorption data is characterized by the frequent occurrence of multiple peaks
corresponding to certain ions, and the thermograms in all experiments reveal four characteristic
desorption patterns, which exhibit the following trends: (i) from 45 to 85°C, some monomer
species undergo almost complete desorption. (ii) Some monomers yield two peaks - one in the
low-temperature range and another at significantly higher temperatures. Additionally, some
monomer ions, associated with certain individual species of the monomer families, occurred at
only very high desorption temperatures, owing possibly to the fragmentation of high-mass
oligomers and dimers. Although less prominent than that observed for monomers, a double peak
occurred for several dimers, whereas for other dimers a single primary desorption peak occurred
at mid to high temperatures (110–170°C). The occurrence of multiple peaks is consistent with the
thermal degradation of extremely low-volatility species that desorb only at temperatures >200°C.
Similar behavior has been observed in previous studies (Holzinger et al., 2010; Lopez-Hilfiker et
al., 2014, 2015; Yatavelli et al., 2012), where the secondary peaks observed during desorption
were attributed to the thermal degradation of very low-volatility aerosol components.
Analysis of the desorption profiles (thermograms) may yield additional information about
the properties of each detected chemical species. The gradual heating of the FIGAERO filter
from 25°C to 200°C resulted in a clear volatility-based separation of species and, for each ion
detected, the desorption temperature corresponding to the maximum signal was identified.
Furthermore, the average desorption temperature of the monomer species was typically lower
than that of their dimer counterparts, which are less volatile. Higher masses (than those
associated with the monomer species) were typically desorbed from the FIGAERO filter at higher



temperatures. An example of this characteristic behavior is shown in the average thermograms
(Fig. 3) of several monomer and dimer ions. In general, compounds evaporating at relatively low
temperatures constituted significant fractions of the gas phase, indicative of monomer
contributions to the aerosol.

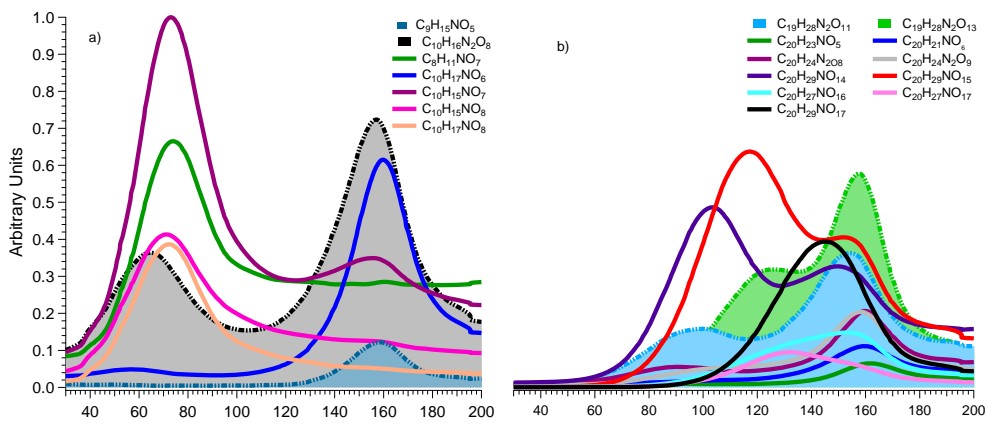


**Figure 3.** Average thermograms (over four desorption cycles) for an $N_2O_5$ ratio of 2.4. Thermograms of ion clusters
of the (a) monomer species ($C_8$–$C_{10}$) and (b) dimer ($C_{19}$–$C_{20}$) species. Ions with double-peak thermogram shape
patterns, consistent with the fragmentation of low-volatility oligomers, are highlighted.

As shown in Fig. 3, each of the detected ion signals reaches at least one local maximum
value. The temperature at which a signal reached the first maximum ($T_{max}$) value was similar
across all experiments (average standard deviation: <10%). Secondary peaks occurred more
frequently for species with a carbon number of 10 or lower, consistent with a degradation-based
contribution. Although the temperature at which the secondary local maximum occurs ($T_{max,2}$)
provides insight into the occurrence of dimerization, the $T_{max}$ value was taken as the true
desorption temperature of each ion.
$T_{max}$ values were identified for each ion in the 196-ion set, and (in general) a positive
correlation ($r^2 = 0.67$) was obtained for the dependence of $T_{max}$ on the molecular mass. A few
monomer, i.e., lower-mass, species ($C \leq 10$) may have formed (via thermal degradation of
higher-MW species) mainly as fragments rather than as primary reaction products and, therefore,
desorbed only at high temperatures. However, four of these ten species ($C_{10}H_{16}O_4$, $C_{10}H_{17}NO_5$,
$C_{10}H_{17}NO_6$, and $C_7H_{10}O_4$) occurred as primary products within the MCM, accounting for (on
average) 69.0 ± 10.8% of the signal detected in the gas phase. This suggests that under the
investigated $N_2O_5$ to limonene ratios, dimer formation from these monomer species (which also
evaporate as fragments during desorption) is highly favored.



The ratio of high-MW species to monomers varied between experiments. At high ratios of
$N_2O_5$ to limonene, the fraction of dimer and oligomer species decreased relative to the total
organic signal, whereas the percentage of high-temperature desorbing monomer species
(fragments) increased. This suggests that absolute dimer formation may have remained the same,
but the monomer signal is over-represented by monomer fragments generated from high-mass,
thermally unstable compounds. The average percentages of particle-phase monomers and dimers
relative to the ratio of reactants across all experiments are shown in Fig. 4. This percentage is
calculated based on the assumption of a common detection sensitivity across all ions; this
assumption may influence the estimated (percentage) contribution of monomers relative to that of
dimers.

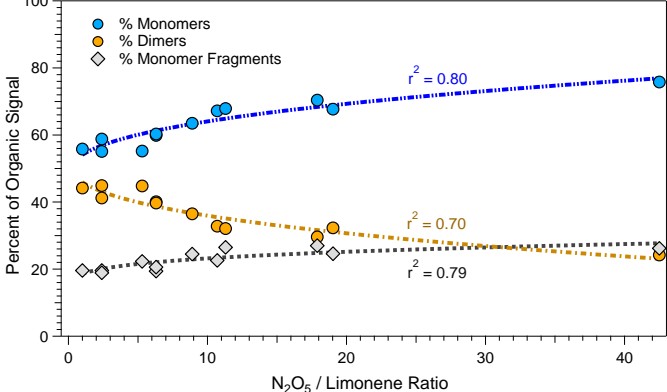

**Figure 4.** Percentage of monomer, dimer, and high-temperature monomer signal (observed during desorption)
relative to the ratio of $N_2O_5$ to limonene injected into the reactor.

### 3.3 Characterization of major SOA products via cluster analysis

Clustering was performed on an ion set consisting of 117 ions, which accounted for >90%
of the total organic signal generated during desorption in all experiments. Ions generating
extremely low signal were excluded to prevent analysis of ions with mis-identified $T_{max}$ values.
However, the occurrence of high-temperature desorbing monomer outliers (described previously)
and the double-peak behavior exhibited by several monomers rendered the mass- and
temperature-based grouping of these ions difficult. To address this issue, duplicate entries,
corresponding to $T_{max}$ and $T_{max,2}$, were assigned to all ions exhibiting double-peak behavior,
allowing the clear separation and analysis of low-mass ions desorbing at temperatures >120°C.
Four and five clusters ($\#_{clust} = \{4, 5\}$), using $T_{max}$, MW, and #C as input, yielded the best
$T_{max}$-based clustering and separation of ions. The use of $n_H$ and $n_O$ as additional input parameters
resulted in partial separation of clusters into groups with similar O/C and H/C ratios, and poor





correlations with respect to $T_{max}$. The average silhouette score obtained for four clusters was
better (0.81 vs. 0.72) than that obtained for five clusters. However, the use of five clusters
allowed for the separation of low-temperature desorbing monomers into two groups with distinct
average $T_{max}$, O/C ratios, and oxidation states. Using more than five clusters resulted in a further
decrease in the quality of *cluster separation, as* measured by the inertia (Eq. 1) and average
silhouette score (Eq. 2). Although the identification of subgroups within each cluster are possible
by increasing $\#_{clust}$, the five main clusters were chosen based on their separation by mass and $T_{max}$
values and to reduce complexity of the interpretation of the resulting clusters with respect to the
chemical composition.
Figure 5a shows the cluster separation on the MW–$T_{max}$ plane. The distribution of
individual cluster members based on oxidation states and #C (Fig. 5b), and the mean MW, $T_{max}$,
O/C, and oxidation state of each cluster (Fig. 5c) are also shown.

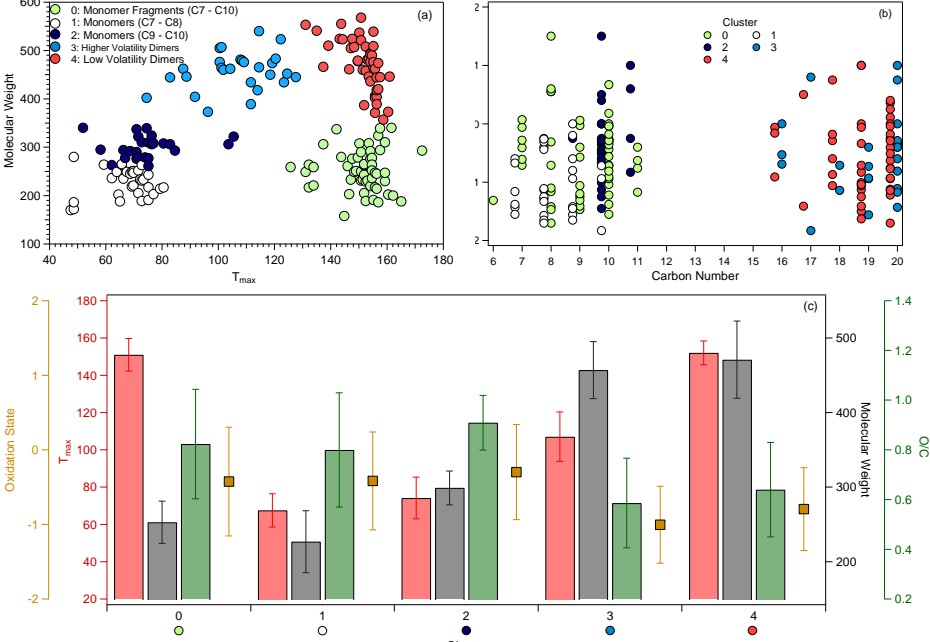


**Figure 5.** Characteristics of the five identified clusters: (a) Desorption temperature of each observed ion in the top
40[th] percentile of ions (identified by their respective desorption signal), color-coded by their corresponding cluster
number, (b) Oxidation state relative to carbon number of all observed ions, colored by their corresponding cluster
membership (for visualization purposes, carbon numbers of groups 0, 2, and 4 are offset), and (c) Average cluster
properties with respect to mass, desorption temperature, and oxidation state.





As Fig. 5 shows, the five clusters are characterized by distinct average MWs and corresponding average $T_{max}$ values. Cluster 0 consists of monomer ions, which are considered fragments of larger, less-volatile molecules, that desorb at high temperatures. The average oxidation state and O/C ratio are similar to those of clusters 1 and 2, which are composed primarily of C7–C9 and C9–C10 monomer ions, respectively. This results from the fact that 87% and 69% of cluster 1 and 2 ions, respectively, have secondary thermogram peaks and $T_{max}$ values, and are represented as members of cluster 0. Ions corresponding to the identified dimers are contained in clusters 3 and 4. The dimers are characterized by two primary desorption regimes, with species that desorb at mid-range temperatures (80–130°C) occurring in cluster 3 and the highest-mass, lowest-volatility ions occurring in cluster 4. Moreover, the distribution of individual cluster members with respect to #C and oxidation state (Fig. 5b) shows that members of high-MW clusters (0, 1, 2) and low-MW clusters (3, 4) reside in separate regimes. The ions in high-MW clusters have a significantly larger number of carbon atoms per molecule and, hence, lower (on average) oxidation states than ions in clusters 0–3. With respect to the most prevalent families listed in Table 2, monomer families m2, m3, and m4 reside exclusively in cluster 2, whereas m5 and m7 reside exclusively in cluster 1. Family members of m1 and m6 were split 20/80% and 75/25% between clusters 1 and 2, respectively. Dimer families d1–d4 occurred predominantly (66–75%) in cluster 4, with the remainder residing in cluster 3. None of the dimer families in Table 2 occurred in clusters 0, 1 or 2.

A positive correlation between the average O/C ratios and $T_{max}$ values was obtained for monomer (clusters 1 and 2) and dimer (clusters 3 and 4) groupings, with more highly oxygenated species desorbing at higher temperatures than their less-oxygenated counterparts. This is also reflected in the overall oxidation state of the clusters ($2 \times O/C - H/C - 5 \times N/C$). However, this trend is observed for only a certain range of masses, as lower MW species had (in general) higher O/C ratios than the dimers. This observation provides some insight into the processes of dimerization that are occurring, indicating the extent to which oxygen is lost during the dimerization process.

**3.4 Mechanisms of dimerization**

Based on the behavior observed in the thermograms for species measured from the limonene SOA in this study, a mechanism of dimer formation, presumably occurring in the particle phase, can be proposed. Dimerization of two monomer species via the loss of one $HNO_3$ molecule may occur during the processes, and $C_{20}H_{29}NO_y$ (y = 10–18) species would be generated from $C_{10}H_{15}NO_x$ (x = 5–9) species. For example, with $HNO_3$ as a leaving group, the mechanism of dimerization between $C_{10}H_{15}NO_6$ and $C_{10}H_{15}NO_8$ (see Reaction 3), would produce the $C_{20}$ dimer species ($C_{20}H_{29}NO_{11}$) that was observed in all experiments. The formation of the observed $C_{19}$ dimer species (e.g., $C_{19}H_{27}O_{15}$) through the combination of, for example, $C_{10}H_{17}NO_7$ and $C_9H_{11}NO_{11}$ monomer species (Reaction 4) is also attributed to this mechanism. Additionally, the occurrence of dimer species with two nitrogen atoms, through the combination of monomers such as $C_{10}H_{16}N_2O_9$ and $C_9H_{13}NO_8$ (Reaction 5), can also be attributed to this dimerization mechanism.



| | | |
|---|---|---|
| 454 | $C_{10}H_{15}NO_6 + C_{10}H_{15}NO_8 \rightarrow C_{20}H_{29}NO_{11} + HNO_3$ | (R3) |
| 455 | $C_{10}H_{17}NO_7 + C_9H_{11}NO_{11} \rightarrow C_{19}H_{27}NO_{15} + HNO_3$ | (R4) |
| 456 | $C_{10}H_{16}N_2O_9 + C_9H_{13}NO_8 \rightarrow C_{19}H_{28}N_2O_{14} + HNO_3$ | (R5) |

The higher O/C ratios of the monomer species, compared with those of the
dimers/oligomers, may also be attributed to the loss of an $HNO_3$ molecule (from the monomer)
during the dimerization process. For example, the two $C_{10}$ reactants in Reaction 3 have O/C ratios
of 0.6 and 0.8 while the product, $C_{20}H_{29}NO_{11}$, has an O/C ratio of 0.55. A similar trend is
observed for Reactions 4 and 5, where the reactants have an average O/C ratio of 0.96 and 0.89,
respectively, and the products have O/C ratios of 0.79 and 0.74, respectively. Due to the loss of
$HNO_3$ during dimerization, dimer fragmentation is expected to yield monomers which differ from
the original (i.e., pre-dimerization) monomers. However, the resulting monomers may also be
associated with aerosol phase products that have secondary desorption peaks. For example, the
fragmentation of $C_{20}H_{29}NO_{11}$ could yield $C_{10}H_{14}O_6 + C_{10}H_{15}NO_5$ or $C_{10}H_{16}O_5 + C_{10}H_{13}NO_6$, and
the fragmentation of $C_{19}H_{27}NO_{15}$ might yield $C_9H_{13}NO_6 + C_{10}H_{14}O_9$ or $C_9H_{13}NO_9 + C_{10}H_{14}O_6$.
Likewise, $C_9H_{13}NO_7 + C_{10}H_{15}NO_7$ or $C_9H_{13}NO_9 + C_{10}H_{15}NO_5$ monomer pairs could be generated
from the thermal degradation of $C_{19}H_{28}N_2O_{14}$.

The fragmentation of dimers may also proceed through multiple channels, thereby
producing several sets of monomer fragments, or the fragmentation of multiple dimers may
produce the same ions. Therefore, attributing the production of a monomer fragment to the
thermal degradation of a specific dimer is difficult, using the current dataset. Large (C > 20)
oligomeric species may contribute to the high-temperature generation of monomer fragment
species. The proposed mechanisms may play only a partial role in the dimerization process
occurring in these experiments. However, they offer a plausible explanation for the occurrence of
multiple observed dimers and the secondary desorption maxima associated with the monomer
constituents.
**4 Conclusions**
High-resolution mass spectrometric data was obtained for both gas and condensed-phase
reaction products resulting from $NO_3$ initiated oxidation of the monoterpene, limonene. The
results revealed that the formation of organic nitrates contributed substantially (89.5 ± 1.4% of
the particulate-phase ion signal) to SOA formation, with dimers constituting a significant fraction
of the particle-phase products. On average, monomers and dimers/oligomers contributed 62.6 ±
7.2 and 37.4 ± 7.3%, respectively, of the particle-phase organic signal detected by the I-CIMS.
Furthermore, many monomers (accounting for 21.6 ± 3.1% of the average organic signal)
desorbed at high temperatures (120°C). The fraction of the signal generated by monomers
increased with increasing $N_2O_5$/limonene ratio (ratio of 42.5 yields a fraction of 75.8%), whereas
the fraction of dimers decreased (to 24.2%). The fraction of the monomer signal resulting from
desorption at high temperatures (≥120ºC) also increased (by 26.2%). Therefore, although the
monomer fraction increased with increasing $N_2O_5$/limonene ratio, this increase in desorption





signal occurred primarily at temperatures above 120ºC, indicative of an increase in the fragmentation of high-MW dimers and oligomers. A large portion (79%) of the monomer thermograms exhibited this bi-modal behavior, with secondary peaks occurring above 120ºC, indicating that the composition of SOA was largely determined by the formation of thermally unstable, low-volatility oligomers.

In total, 196 individual organic ions were detected during desorption. However, the total measured organic signal was generated mainly by 52 (i.e., 76%) of these ions, which constituted the 75th percentile of the monomer and dimer signals. Over half of the signal emanated from the top 90th percentile, which comprised a small subset of only 20 species, of the total number of ions. These 20 species (with nine listed as major products in the MCM) constituted the major particle-phase products formed via the reaction of $N_2O_5$ and limonene under the conditions employed in this study. The non-listed species were either dimer species or more highly oxygenated, nitrated analogs of known major products, which are notoriously hard to describe via standard gas-phase mechanisms.

Cluster analysis revealed two monomer groups, two dimer groups and, a separate group containing monomer ions that exhibited secondary desorption peaks occurring at temperatures ≥150°C. Each group was characterized by a distinct average MW and desorption temperature ($T_{max}$). The 2 identified clusters in the monomer and dimer sub-classes differ in oxidation state and O/C ratios, with increasing O/C corresponding to higher $T_{max}$ values. Based on the analysis of these monomer and dimer groups, a mechanism (Reactions 3–5) was proposed for the formation of the observed dimers from the monomers.

Using a combination of cluster analysis and thermal properties derived from Figaero-CIMS measurements may provide some means of reducing the complexity associated with the description of SOA formation processes. The investigated reaction system constitutes only one of many systems, but could be used as an example of the evaluation required for this type of information derived from high-resolution MS. The results revealed that, analogous to products from ozonolysis and •OH-induced oxidation, the organic nitrates produced in the nighttime chemistry of biogenic compounds comprise a multi-component mixture that contributes to ambient SOA. Thus, the aerosol species detected here could be included in modeling studies with the aim of explaining scenarios where SOA formation rates are under-predicted. Furthermore, the numerous products resulting from $N_2O_5$ oxidation of limonene, which were identified and grouped based on thermal properties, could be candidates for identification in ambient air masses dominated by nocturnal limonene chemistry.

**Competing interests**

The authors declare that they have no conflict of interest.





## Acknowledgment

The research presented is a contribution to the Swedish strategic research area ModElling the
Regional and Global Earth system, MERGE. This work was supported by the Swedish Research
Council (grant numbers 2015-04123; 2014-05332; 2013-06917), Formas (grant number 2015-
1537)

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
