# Peer review of "Characterization of organic nitrate constituents of secondary organic aerosol"

_Atmospheric Chemistry and Physics, 2017_

## Referee Comment (RC1) · Anonymous Referee #1 · 18 Jul 2017

This manuscript describes data collected from a FIGAERO-CIMS during laboratory studies using the flow reactor GFROST. Focus is given on the major organic nitrate constituents of secondary organic aerosol from the NO3 oxidation of limonene. A cluster-analysis method is used in order to distinguish the groups of ions based on their thermodynamic properties and molecular formula information. Mechanisms of dimer formation are suggested based on the grouping performed. The paper is suitable for publication in ACP. My suggestions below are mainly to clarify the presentation

[Figure]

and context of the results.

Specific comments

1. Recent studies have shown that fragmentation due to thermal dissociation in the FIGAERO-CIMS could strongly affect the chemical formula attribution (Stark et al., 2017). It would be beneficial if a comparison of the different methods of assigning a chemical formula to the major ions could be performed. If not, then the possible uncertainties introduced should be further discussed.

2. In line 98 it is mentioned that one of the main focuses of this paper is to determine the molecular formula of species that could contribute significantly to SOA formation. Although the uncertainty of calculating the mass concentration using FIGAERO-CIMS would be high it could still indicate whether the compounds measured are indeed a large fraction of the overall mass as has been done from Isaacman-VanWertz et al. (2017).

3. FIGAERO-CIMS collects both the gas and particle phase compounds on the filter. More information on the gas phase compounds detected would bring light concerning the extent of possible gas phase "interference" during desorption. When the signal of the compounds in the gas phase is high then their contribution to the aerosol during desorption could increase more. Have there been any checks on the contribution of gas phase signal during desorption? How much is it expected to be in the monomer and how much in the dimer range? A figure to illustrate this effect in the supplementary would be very informative. For example, in lines 346-348 this could also be due to gas phase compounds collected on the filter that undergo evaporation and thus contribute to the aerosol. What is the gas phase concentration of these compounds compared to the aerosol?

4. In Figure 3 for many of the compounds there is a residual signal above zero. How would that affect the signal of the next desorption? When FIGAERO reaches stable conditions what does that exactly mean? It would be fruitful for the reader if background

measurements could be provided during these experiments on the Supplementary material. An additional figure to support the stability of the thermograms would be very instructive. For example the average cumulative signal (with error bars indicating the standard deviation of the average) vs temperature, for each compound, with each compound indicated with a different color would be a suggested way.

5. What is the atmospheric relevance with the mixing ratio of N2O5 used? What is the NO3 expected mixing ratio in the system? A discussion would better inform the reader.

6. In Figure 4 what is the fit function used? What is the information we gain from this fit choice? Further discussion in the manuscript would clarify these questions. This figure doesn't provide all the data points seen in Table 1. The high N2O5/limonene ratio of 113.3 is missing. Could that be included or provided in the supplementary? Is this point included in the fit function applied now? Finally, error bars for the y-axis are missing and the fit function should be applied by taking into account this error.

7. In section 2.3 was it both gas and particle phase data used as identified species to compare to the MCM? A sentence to make clear that this work is focused on the particulate phase and not the gas phase would better direct the reader at this point.

8. It would be beneficial for the reader if the discussion in section 3.4 was extended on how the suggested mechanisms can be directly inferred from the grouping technique and the volatility of the compounds.

Technical comments

Line 47. A broader view of organic nitrates is given by Kiendler-Scharr (2016).

Line 154. Units are missing.

Line 167. Delete one dot.

Line 234. Recent findings suggest that fragmentation due to thermal dissociation occurs in systems like the FIGAERO-CIMS (Stark et al., 2017). See comment 1.

Line 324-337. The authors provide four characteristic desorption patterns and the numbering stops at two.

Line 361-362. It would be more informative to add the positive correlation in the supplement as a figure.

Line 387. Please define what extremely low signal would be as a value.

Line 428. Should be 0-2?

Table 1. Sorting of the lists would make the table more readable. Sorting the N2O5 from low to high and within each N2O5 group sorting the limonene from low to high would be one way of performing the sorting. Adding the expected NO3 concentration would be fruitful. Errors for the average SOA mass measured from an SMPS are missing. An additional column of the mass FIGAERO-CIMS could detect (with its much higher uncertainty), as discussed above, would be an indicator of how much of the overall SOA mass is measured in this system.

Table 2. What is the information we gain from the (N/C)x10? Based on all uncertainties the temperature precision could be rounded. Error of the average contribution is missing.

Figure 3. Certain double peak compounds are not highlighted like C10H15NO7, C20H29NO5, C20H24N2O8 etc. The C20H24N2O8 is not written correctly in the annotation. Since there are more compounds that have double peaks plotted it would be clearer to include the double peak compounds with dash lines and avoid highlighting.

Figure 5. It improves the figure if (a) and (b) have the same annotation introduced on the right side outside both figures once. This way you avoid the change in font size that is seen for annotation from Figure (a). For Figure (b) the oxidation state is not mentioned in the axis and the range is not going to minus when it is below zero. It would be beneficial if Figure (c) was separated in two graphs. On the left side a figure of the MW (left-axis) and Tmax (right-axis) and on the right side a figure of the oxidation

state (left-axis) and the O/C (right-axis). Box-and-whiskers instead of bars and markers would provide more information on the dataset. This would also show more clearly the temperature increase that is suggested to correlate to the O/C increase. Finally the colors of Figure (c) are similar to the colors of the clusters thus confusing the reader.

References

Isaacman-VanWertz, G., P. Massoli, R. E. O'Brien, J. B. Nowak, M. R. Canagaratna, J. T. Jayne, D. R. Worsnop, L. Su, D. A. Knopf, P. K. Misztal, C. Arata, A. H. Goldstein, and J. H. Kroll: Using advanced mass spectrometry techniques to fully characterize atmospheric organic carbon: current capabilities and remaining gaps, Faraday Discussions, doi:10.1039/C7FD00021A, 2017.

Kiendler-Scharr, A.: Ubiquity of organic nitrates from nighttime chemistry in the European submicron aerosol, Geophys. Res. Lett., 43, 7735–7744, doi:10.1002/, 2016.

Stark, H., R. L. N. Yatavelli, S. L. Thompson, H. Kang, J. E. Krechmer, J. R. Kimmel, B. B. Palm, W. Hu, P. L. Hayes, D. A. Day, P. Campuzano-Jost, M. R. Canagaratna, J. T. Jayne, D. R. Worsnop, and J. L. Jimenez: Impact of Thermal Decomposition on Thermal Desorption Instruments: Advantage of Thermogram Analysis for Quantifying Volatility Distributions of Organic Species, Environ Sci Technol, doi:10.1021/acs.est.7b00160, 2017.

---

## Referee Comment (RC2) · Anonymous Referee #2 · 24 Jul 2017

General comments: This paper presents novel flow tube measurements of the chemical composition of SOA produced from NO3 + limonene, with the potential to provide valuable new mechanistic clues and making the (as far as I know) new proposal that dimer formation reactions may be accompanied by HNO3 loss, as a possible explanation for the C20 compounds observed with only a single N. The authors also describe observed thermal desporption profiles that hint at monomers coming from dimer dissociation in the instrument inlet, an important caution to other researchers using this

technique. For these reasons, I think this manuscript will be a valuable contribution to the atmospheric chemistry literature, but it needs substantially more work before it is ready for publication. First, in many places I found the writing confusing and had a difficult time understanding what the authors were saying – I think this draft was at least one round of serious editting away from being ready to submit. I'll flag the sentences I found most confusing below, but after edits, I urge the authors to also find another outside reader to go through the entire manuscript. Second, and more importantly, the analysis of this rich dataset feels incomplete. At present, the paper really just presents FIGAERO MS and thermogram results, cluster analysis, and the HNO3 loss mechanism idea, without much support beyond the chemical formulae observed. I make the following broad suggestions for the authors to further this analysis before resubmitting for publication:

(1) present a better foundation for the idea of HNO3 loss: show a proposed structural mechanism for how this would occur, cite additional literature on the enegetics of such a reaction and of any other experiments that made observations consistent with this proposal, and try to produce an (even coarse) nitrogen budget from your experiments to check if it supports HNO3 loss. How much of the N2O5 you injected didn't show up in gas or aerosol phase CIMS products? is any change in this budget over different experiments consistent with the amount of the products you hypothesize to come from this type of reaction?

(2) The cluster analysis should also be analyzed and discussed in greater detail: thus far, it seems to serve only to highlight the same two "groups" as the MS alone would have – the monomer region (in 3 factors) and the dimer region ( in 2 factors). You mention the families in each cluster and have generic chemical formulae for them. Can you propose structures / reasons these would be different? Are they potentially from oxidation at the different double bonds (what would you expect that to look like?) or form RO2 + RO2 reactions vs. RO2 + NO3 reactions? Why is cluster 0 basically spread across the O;C and MW space of clusters 1 and 2 – why is it nevertheless a

separate cluster? Is there any proposed mechanism that would get you this, or could it be that cluster 0 are the fragments of dimers while 1 and 2 are straight monomers? Or, some permutation of this?

(3) It seems you should also be able to do more with the fact that some monomers have double peaks and some don't – can you correlate this to the cluster analysis somehow, or otherwise interpret it mechanistically?

(4) You mention that some observed formulae are in MCM and others are not. Do more with this. Are there any structures predicted to be major products in MCM that you don't observe? I would suggest to show a modeled output of MCM (just a box model) for your expt. conditions. Are the major few predicted products all those that you observe, or are you just observing one product channel / some at random / etc.? You have the data now to truly test the MCM (and you allude to it), and I was disappointed to see that you didn't report a true comparison. For the major formulae that you observed that aren't in MCM, too, I'd like to see more analsysis. You say in your conclusions that these should be included in models, but you haven't told us what they might be. Propose some structures / mechanistic origins / something based on MCM to help guide how your novel observations might be incorporated.

(5) Suggest to read this recent NO3 + limonene paper from the Ng group: http://pubs.acs.org/doi/abs/10.1021/acs.est.7b01460 and include comparisons to these results in your updated draft. Can you make any quantitative estimate of SOA yields to compare to Boyd et al.'s (very high!!) SOA yields from NO3 + limonene, or comment on them based on your mechanism? Are >100% yields consistent with your proposed dimerization mechanism, or inconsistent? Please discuss.

Specific editorial suggestions:

in title: "nitrate-radical-initiated" should be "nitrate radical-initiated", and elsewhere in text

line 11 "reaction of NO3 with limonene"

line 14 "identity and volatility of the most"

Sentence line 17-19 is confusing. How about: "The observed products were compared to those in the Master Chemical Mechanism (MCM) limonene mechanism, and many non-listed species were identified."

line 39-40: "The oxidation of VOCs by"

line 49: find some more recent refs to add to cite list

lines 55-58: seems odd not to have the NO3 + R formation of organonitrates listed here, since this is the one you focus on

line 97 "molecaulr formulae of major nitrate species produced"

line 100 "(based on the Master Chemical..."

line 154 "1.4 g cm-3 (Hallquist..." { units}

line 167: mention typical [NO3] here too, and comment on whether RO2 + RO2 or RO2 + NO3 reactions should be dominant?

in caption for Table 1, mention hos you know the [N2O5]. Can you use these values to get an approximate SOA yield?

line 210 "silhouette score, s(i) (Rousseuw...."

around line 236-240: "regions" is confusing – define as the deignated mass ranges if you'll use for further discussion.

237-238 "These regions corresponded to aerosol samples" : what does this mean? They were always enhanced in aerosol samples relative to gas, or ...?

line 245: you refer to specific exptal condition of ratio = 2.4. This sounds contradictory to your statement that the regions "always" correspond to aerosol (I interpreted this as

meaning under all ratios of reagents)

line 252: you call C11 a dimer? Also, the carbon number ranges here don't correspond to your shaded regions in Fig. 2, shouldn't they?

line 257: now you're introducing a new designation, the "first" region for what you've called the "monomer" region before (I assume these mean the same thing). I urge you to define a term that refers to these regions at the beginning of this section and stick with it. Even better: label it on Fig. 2.

line 272-273: confused by "at least modestly" ?? wouldn't this just be the remaining 56.5%? I wouldn't call that modestly. Or I don't understand to what you refer here.

line 274-275: confused by "the monomers contain": many? all? on average?

line 276: "Progressively more oxygenated monomers of the general . . ."

line 277: omit the paranthetical after 5—9, this is clear.

line 283: how would you know a structure is a PAN? Unless you show a mechanism that would get to those species, I'd just leave this discussion as PNs only, they would all readily dissociate.

line 292: where did the range of formulae you cite for ELVOC come from? Is this your defition, or does it come from Donahue? (similar question for dimer formulae on line 310)

line 299: confused by "using $\alpha$-pinene." using it as what? Nighttime SOA at SOAS is more likely to be from $\beta$-pinene, since its NO3 SOA yields are higher. Not sure what you mean here by invoking $\alpha$-pinene.

lines 299-303: rework into one sentence; repetitive; confusing. (Maybe because of trying to tie back to the a-pin reference? which you could just remove)

line 304-35: not clear what this sentence means.

line 310-311: parenthetical seems internally contradictory: "only" in aerosol phase, but also "slightly about background" in the gas phase?

line 314 "ion families (defined as groups of molecular formulae with only the number of O atoms varying) to the total . . ." {correct?}

Table 2: the large ranges but with very precise end points for the Tmax reported is weird. If you really think only the range is interesting, truncate sig figs. If as I suspect the actual list of Tmax values for each member of the family (there are always 6 or fewer, so not too crazy) might be interested to aide your interpretation, why not list them instead, on the order of x value to people can see which correspond? then you can discuss more about which Tmax's appear across monomers / dimers and do some analysis / inference about connectedness from those!

line 328 "Additionally, (iii) some. . ."

line 346-348: how is this indicative of monomer contributions? do you mean indicative of LOWER monomere contributions to aerosol?

Fig 3: why are some traces shaded to zero? describe/ discuss or don't have the difference. in caption, "N2O5 ratio" should be "N2O5 to limonene ratio" , right?

lines 362 / 365: do the "few" and the "ten species" refer to the same subset? This para is confusing.

lines 367-369: not sure how / why this suggests favored dimer formation – confusing

line 370: another new designation, "high-MW"! dimer/oligomer? be consistent in how you refer to the different groups

line 373: cite Figure 4

line 374: do you mean "the monomer signal at higher N2O5 to limonene ratio"? If so, say so . If not, clarify what you mean

line 375-376: "The average ... Fig. 4" : remove this sentence, not clear & not helpful, and can cite Fig. 4 above instead.

line 407: define oxidation state here, where you first mention it, instead of later where you currently define it

Fig. 5: put the labels for mass, desorption temperature, O/C and oxidation state in the same order (1) on the plot, (2) in the caption (and mention all 4!), and put the y axes in the same order, to make life easier on your readers. Also, you show S.D. on the plot (I guess? or, what are the error bars?) – mention this in caption.

line 421 do you mean "and are also represented as members of cluster 0"?  i.e., the ions are members of both clusters? Clarify.

lines 424-428: Could an alternative rationale for this difference in oxidation state simply be that monomers, because of their small carbon chain length, need more oxidized functional groups to condense, while dimers are so big they'll condense even with less oxidation? Since you are looking only at the aerosol phase here, this pattern could be skewed by differing volatility.

line 437: oxidation state is defined here, should be at first instance of it earlier in text.

line 438: @ "certain range of masses", state the range

line 440: same comment as lines 424-428.

line 445 "during the process, and"

line 446: as mentioned in general comments above, I think it would be good to include a structural diagram of the HNO3 leaving reaction.

lines 450 & 452: since this relied on having C9 monomers, it would be useful also to explain how these are formed in the mechanism leading up to dimerization.

line 453: at the end of this discussion, I think it would also be good to at least mention

any alternative possible pathway that could make these dimer products – is it really only possible with HNO3 loss, or could you get it some other way? Then, perhaps the nitrogen balance or other evidence can help you bolster your hypothesis that the HNO3 loss is the more likely route.

lines 463-464: "dimer fragmentation . . . monomers." is a confusing sentence

lines 465-469: can't you demonstrate this more conclusively by looking at specific examples of masses with and without double peaks, and possibly matching up the Tmax's, and thus identify the subset of monomers that are also dissociation products and possibly connect them to precursors? Even better – if relative amounts change with different reaction conditions, can you track them rising and falling together? This dataset would seem to have lots of potential to demonstrate these actually linkages, not just make vague statements and what might / could yield what else upon fragmentation.

line 480: "obtained for both gas and condensed-phase" – since you don't discuss gas phase data in this paper, omit, or cite to the companion paper that does study gas phase?

line 484-485: lots of sig figs on these percentages considering the error bars – better to say 63 +/- 7 and 37 +/- 7 %? And perhaps put a few numbers in the abstract?

line 494-496: this has me wondering whether you can learn anything from the relative intensities of the two peaks? would this pattern support that SOA is "largely determined" by low-volatility oligomers?

line 504-505: as mentioned in the general comments, I think you should at least do some discussion of the non-listed products and what they might be.

lines 509-512: are all the hypothesized HNO3-loss dimers in one cluster and others in another? Or, how else can you use the cluster analysis to learn something related to your mechanism speculation?

line 513: "FIGAERO" to be consistent with how you write it above

line 514: "may provide some means of reducing the complexity..." see above general comments. I hope you can work get a bit more out of this.

line 520-521: as mentioned above, hard to include species in modeling studiest if you haven't even suggested what you think they might be.
* * *

---

## Referee Comment (RC3) · Anonymous Referee #3 · 22 Aug 2017

This work investigated organic nitrate formation from NO3 oxidation of limonene. Experiments were conducted using different N2O5/limonene ratios. Speciated gas and particle phase organic nitrates were measured by the FIGAERO-HR-ToF-CIMS. Cluster analysis of the desorption temperatures of organic nitrate species resulted in five clusters; the relationships between O/C, OS, MW, etc of these clusters were discussed. Formation of dimers was observed and reaction mechanisms for dimer formation were proposed.

[Figure]

This is an interesting study and the manuscript is generally well-written and easy to follow. This study will be of interest to the greater atmospheric community. My main comments are 1) while the experiments were conducted over a range of N2O5/limonene ratios, the authors shall provide more context to this experimental design. Also, the results from experiments with different N2O5/limonene ratios need to be more extensively and clearly discussed. 2) I have some concerns regarding the discussion of the results shown in Fig. 5, please see details below. 3) There are a number of recent studies on nitrate radical oxidation of biogenic hydrocarbons, it would be appropriate that these studies are referenced in the manuscript to reflect the current state of knowledge.

Overall, I recommend publication in ACP once these comments are addressed. Most comments are mainly to improve clarity of the manuscript.

Main Comments

1. Page 5 line 158. Are potential impurities (e.g., NO2 and HNO3) in the N2O5 synthesized measured and quantified? Please make this clear in the manuscript.

2. Page 5 line 168. What is the reason for performing experiments with different N2O5/limonene ratios? Please provide more context here.

3. Page 6 Table 1.

a. With these N2O5/limonene ratios, are all limonene (and both double bonds?) reacted away? Please clarify and change the "limonene" in the table to either "reacted limonene" or "initial limonene".

b. What is the RO2 reaction regime in these experiments? RO2 + NO3? RO2 + RO2?

c. Can SOA yields be quantitatively calculated from the values in the table? If the "limonene" is reacted limonene, the SOA yields appear to be very low compared to previous studies by Fry et al. (ACP, 2009), Fry et al. (ACP, 2011), and Boyd et al. (ES&T, 2017). Please discuss the results from this study in the context of these prior studies. Also, do the data shown in Table 1 follow a typical Odum 2- product yield

curve?

4. Page 8 line 238. It was noted that "….the relative signal intensities varied with the amount of limonene and N2O5 present in the system". I think the authors are referring to Fig. 4? Please add the figure number to the sentence to help guide the readers if this is the case.

5. Page 8 line 260. It is not immediately clear what these species are without diving into the entire MCM mechanisms. The authors should at least include the formation mechanisms of these major ions in the SI to help guide the readers. Also, it would be helpful to propose mechanisms for the major species that were observed in this study but are not in MCM. On the related note, Boyd et al. (ES&T, 2017) recently expanded the limonene + NO3 mechanisms in MCM. It might be worthwhile to evaluate if some species detected in this study are covered in the expanded mechanism in Boyd et al.

6. Page 9 line 280. What are some of the mechanisms for limonene and its oxidation products to react with NO2 and HNO3?

7. Page 10 line 293. Nah et al. (ES&T 2016) also measured a large suite of highly oxygenated organic nitrates from NO3 oxidation of a-pinene and b-pinene in laboratory experiments, using the FIGAERO-HR-ToF-CIMS. Many of those are also observed in Lee et al. (PNAS, 2016).

8. Page 10 line 299. It is noted that "The similarity with ions from the NO3-initiated limonene oxidation further emphasizes the importance of monoterpenes as precursors of organic nitrates". It would be informative if the authors indicate in the Table in SI (the ion list) regarding which ions have also been observed in the ambient (Lee et al.) and other monoterpene experiments (Nah et al).

9. Page 10 line 301. Rollins et al. (Science, 2012) discussed the importance of limonene + NO3 in a field study.

10. Page 12 line 362. It was noted that there is a positive correlation ($R^2 = 0.67$)

between Tmax and molecular mass. Are the authors referring to Fig 5a? If so, it does not look like the overall correlation is that good? Please clarify.

11. Page 13 line 370-379. (this is also related to comment #2 above). It is not clear to me how the higher N2O5/limonene experiments lead to formation of more thermally unstable mechanisms. Please explain. More discussions are needed here, to provide context to why experiments are conducted with different N2O5/limonene in the first place and why/how the resulting compositions are different.

12. Page 13 Figure 4. What is the function used for the fit? Is this just to guide the eye or there is a fundamental reason for such a dependence?

13. Page 15, discussions of Figure 5. The authors attempted to discuss the relationships between O/C, OS, and MW, etc. However, within uncertainties, there do not seem to be significant differences in the O/C and OS values for all clusters. Hence, this discussion needs to be revised.

a. (related to comments # 2 and 11 above). In Figure 5, will any specific patterns emerge if the authors only look at the data from experiment of a particular N2O5/limonene ratio?

b. Line 418. It was noted that the O/C of cluster 0 is similar to clusters 1 and 2. However, within the uncertainties, the O/C ratios of all clusters are almost the same.

c. Line 426, should the high-MW clusters be (3,4)? And the low-MW clusters be (0, 1, 2)?

d. Line 427. It was noted that the ions in the high-MW clusters have a lower OS than ions in clusters 0-3. Firstly, should "0-3" be "0-2"? Secondly, it does not look like the high-MW clusters have a lower OS. Within uncertainties, the OS values appear to be the same for all clusters.

e. Line 434. It was noted that a positive correlation exists between O/C and Tmax. It is not clear how this is case from the data shown in Fig. 5c. Please provide a figure and

show the R2 value.

Minor Comments

1. Page 2 line 44. Would be appropriate to reference Ng et al. (ACP, 2017).

2. Page 2 line 46. Would be appropriate to also reference Day et al. (AE, 2010); Fry et al. (ACP, 2013); Xu et al. (PNAS, 2015); Xu et al. (ACP, 2015); Boyd et al. (ACP, 2015); Kiendler-Scharr et al. (GRL, 2016); Nah et al. (ES&T, 2016).

3. Page 2 line 60. Delete "M" in front of Hallquist.

4. Page 3 line 77. Boyd et al. (ES&T, 2017) recently investigated SOA formation from NO3 oxidation of limonene.

5. Page 3 line 85. Would also be relevant to cite Xu et al. (PNAS, 2015), Xu et al. (ACP, 2015), and Rollins et al. (Science, 2012). Out of all the references listed in the manuscript and these few ones I mentioned here, Rollins et al. will likely be the most relevant to this study due to the relatively higher concentrations of limonene at Bakersfield.

6. Page 14 Figure 5. Missing y-axis label for 5b?

---

## Author Comment (AC1) · 15 Dec 2017

We thank the reviewers for the helpful comments! We will now take the benefit from those in improving the manuscript. A point by point response (in black) to the reviewers' comments (in blue) will follow.

**Response to Anonymous Referee #1**

This manuscript describes data collected from a FIGAERO-CIMS during laboratory studies using the flow reactor GFROST. Focus is given on the major organic nitrate constituents of secondary organic aerosol from the NO3 oxidation of limonene. A cluster-analysis method is used in order to distinguish the groups of ions based on their thermodynamic properties and molecular formula information. Mechanisms of dimer formation are suggested based on the grouping performed. The paper is suitable for publication in ACP. My suggestions below are mainly to clarify the presentation

**Reply:** Thanks for the suggestions to clarify the presentations.

Specific comments 1. Recent studies have shown that fragmentation due to thermal dissociation in the FIGAERO-CIMS could strongly affect the chemical formula attribution (Stark et al., 2017). It would be beneficial if a comparison of the different methods of assigning a chemical formula to the major ions could be performed. If not, then the possible uncertainties introduced should be further discussed.

**Reply:** Yes, we should more clearly acknowledge the possibilities for decomposition but would rather be brief and refer to the findings in the Stark et al paper

**Action:** Reference added with discussion on uncertainties.

2. In line 98 it is mentioned that one of the main focuses of this paper is to determine the molecular formula of species that could contribute significantly to SOA formation. Although the uncertainty of calculating the mass concentration using FIGAERO-CIMS would be high it could still indicate whether the compounds measured are indeed a large fraction of the overall mass as has been done from Isaacman-VanWertz et al. (2017).

**Reply:** Assuming common sensitivities for all compounds would enable the reader to find the major contributors to SOA mass in the list of compounds (Table 1 supplemental). The application of various sensitivities for I-CIMS is to date very uncertain and discussed in the literature, e.g. Isaacman-VanWertz et al. (2017).

**Action:** The Isaacman-VanWertz et al. (2017) has been added as reference for current state where they summarising methods to derive concentrations from I-CIMS . In addition, a short notice on the issue brought up by the referee with reference to Table 1 has been added.

3. FIGAERO-CIMS collects both the gas and particle phase compounds on the filter. More information on the gas phase compounds detected would bring light concerning the extent of possible gas phase "interference" during desorption. When the signal of the compounds in the gas phase is high then their contribution to the aerosol during desorption could increase more. Have there been any checks on the contribution of gas phase signal during desorption? How much is it expected to be in the monomer and how much in the dimer range? A figure to illustrate this effect in the supplementary

would be very informative. For example, in lines 346-348 this could also be due to gas phase compounds collected on the filter that undergo evaporation and thus contribute to the aerosol. What is the gas phase concentration of these compounds compared to the aerosol?

**Reply**: The methods and potential artifacts have been described in previous publications. Generally, there are limiting gas-phase contaminations of the condensed phase evaporation. Yes, the gas-phase has also been measured and could be presented to illustrate the ratio between gas and particle phase concentrations.

**Action:** Two figures illustrating partition (ratio between the gas and the particle phase) for ions in monomer and dimer region has been added in supplemental.

4. In Figure 3 for many of the compounds there is a residual signal above zero. How would that affect the signal of the next desorption? When FIGAERO reaches stable conditions what does that exactly mean? It would be fruitful for the reader if background measurements could be provided during these experiments on the Supplementary material. An additional figure to support the stability of the thermograms would be very instructive. For example the average cumulative signal (with error bars indicating the standard deviation of the average) vs temperature, for each compound, with each compound indicated with a different color would be a suggested way.

**Reply:**  Usually several cycles are repeated and an average of three desorptions are used for the results

**Action:** Examples of three consecutive desorptions are now presented in supplemental

5. What is the atmospheric relevance with the mixing ratio of N2O5 used? What is the NO3 expected mixing ratio in the system? A discussion would better inform the reader.

**Reply:** Obviously, the mixing ratio of $N_2O_5$ is higher than ambient to provide enough amount of limonene reacting during the time spent in the flow reactor. At a ratio of 1:1 we expect $NO_3$ to primarily react with the exocyclic double bond, representing the typical case producing primary products, while increasing the ratio will add more possibilities for reactions also to the endocyclic double bond reflecting secondary atmospheric chemistry.

**Action:** Short discussion on our view on this has been added in the manuscript

6. In Figure 4 what is the fit function used? What is the information we gain from this fit choice? Further discussion in the manuscript would clarify these questions. This figure doesn't provide all the data points seen in Table 1. The high N2O5/limonene ratio of 113.3 is missing. Could that be included or provided in the supplementary? Is this point included in the fit function applied now? Finally, error bars for the y-axis are missing and the fit function should be applied by taking into account this error.

**Reply:** yes, we agree the line is mostly for guidance of the eye and should be removed. The data point at 113.3 was not illustrated as it would extend the x-axes too far.

**Action:** The r2 has been removed and the missing data point at 113.3 is now noted in the footnote.

7. In section 2.3 was it both gas and particle phase data used as identified species to compare to the MCM? A sentence to make clear that this work is focused on the particulate phase and not the gas phase would better direct the reader at this point.

**Action:** A sentence has been added to direct the reader on this point

8. It would be beneficial for the reader if the discussion in section 3.4 was extended on how the suggested mechanisms can be directly inferred from the grouping technique and the volatility of the compounds.

**Reply:** The lower O/C ratios and loss of one nitrogen compound for some of the dimers while exhibiting a low volatility would support the mechanism suggested. (loss of oxygen(s) and nitrogen in dimerisation)

**Action:** This info has now been clarified.

**Technical comments**

Line 47. A broader view of organic nitrates is given by Kiendler-Scharr (2016).

**Action:** Reference added

Line 154. Units are missing.

**Action:** Units of g cm$^{-3}$ have been added

Line 167. Delete one dot.

**Action:** Full stop deleted

Line 234. Recent findings suggest that fragmentation due to thermal dissociation occurs in systems like the FIGAERO-CIMS (Stark et al., 2017). See comment 1.

**Action:** The following text has been added to acknowledge the work of Stark *et al.* (2017) and we have considered for possibilities that ions may fragment upon desorption "Stark *et* al. (2017) have recently shown that fragmentation during the desorption can occur within the FIGAERO, although we find no fragmentation here which suggests that the identified ions were products of fragmentation of larger molecules, with exception of one group of compounds in the cluster analysis"

Line 324-337. The authors provide four characteristic desorption patterns and the numbering stops at two.

**Action:** This has now been clarified in the text with the addition of extra numbering in the text

Line 361-362. It would be more informative to add the positive correlation in the supplement as a figure.

**Action:** A Figure on Tmax vs Mw has been added in supplementary

Line 387. Please define what extremely low signal would be as a value.

**Action:** A definition has now been added.

Line 428. Should be 0-2?

**Action:** Changed to 0-2 in text

Table 1. Sorting of the lists would make the table more readable. Sorting the N2O5 from low to high and within each N2O5 group sorting the limonene from low to high would be one way of performing the sorting. Adding the expected NO3 concentration would be fruitful. Errors for the average SOA mass measured from an SMPS are missing. An additional column of the mass FIGAERO-CIMS could detect (with its much higher uncertainty), as discussed above, would be an indicator of how much of the overall SOA mass is measured in this system.

**Action:** The Table has been ordered as suggested by the reviewer. We did not measure the actual $NO_3$ concentrations and the total mass from FIGAERO-CIMS is rather uncertain. However, the standard deviation of the mean SOA mass is now included in the Table.

Table 2. What is the information we gain from the (N/C)x10? Based on all uncertainties the temperature precision could be rounded. Error of the average contribution is missing.

**Action:** The column displaying N/Cx10 has been removed. The temperatures have been rounded to the nearest degree. Errors of the average contributions are now included.

Figure 3. Certain double peak compounds are not highlighted like C10H15NO7, C20H29NO5, C20H24N2O8 etc. The C20H24N2O8 is not written correctly in the annotation. Since there are more compounds that have double peaks plotted it would be clearer to include the double peak compounds with dash lines and avoid highlighting.

**Action:** The highlighting has been removed and the mis-formatted compound name ($C_{20}H_{24}N_2O_8$) has been corrected.

Figure 5. It improves the figure if (a) and (b) have the same annotation introduced on the right side outside both figures once. This way you avoid the change in font size that is seen for annotation from Figure (a). For Figure (b) the oxidation state is not mentioned in the axis and the range is not going to minus when it is below zero. It would be beneficial if Figure (c) was separated in two graphs. On the left side a figure of the MW (left-axis) and Tmax (right-axis) and on the right side a figure of the oxidation state (left-axis) and the O/C (right-axis). Box-and-whiskers instead of bars and markers would provide more information on the dataset. This would also show more clearly the temperature increase that is suggested to correlate to the O/C increase. Finally the colors of Figure (c) are similar to the colors of the clusters thus confusing the reader.

**Action:** Changed the plot to a 4-panel plot by separating the bottom graph. The other minor formatting issues were also fixed, like the single legend for the top plots.

References Isaacman-VanWertz, G., P. Massoli, R. E. O'Brien, J. B. Nowak, M. R. Canagaratna, J. T. Jayne, D. R. Worsnop, L. Su, D. A. Knopf, P. K. Misztal, C. Arata, A. H. Goldstein, and J. H. Kroll: Using advanced mass spectrometry techniques to fully characterize atmospheric organic carbon: current capabilities and remaining gaps, Faraday Discussions, doi:10.1039/C7FD00021A, 2017.

Kiendler-Scharr, A.: Ubiquity of organic nitrates from nighttime chemistry in the European submicron aerosol, Geophys. Res. Lett., 43, 7735–7744, doi:10.1002/, 2016.

Stark, H., R. L. N. Yatavelli, S. L. Thompson, H. Kang, J. E. Krechmer, J. R. Kimmel, B. B. Palm, W. Hu, P. L. Hayes, D. A. Day, P. Campuzano-Jost, M. R. Canagaratna, J. T. Jayne, D. R. Worsnop, and J. L. Jimenez: c, Environ Sci Technol, doi:10.1021/acs.est.7b00160, 2017.

**Response to Anonymous Referee #2**

General comments: This paper presents novel flow tube measurements of the chemical composition of SOA produced from NO3 + limonene, with the potential to provide valuable new mechanistic clues and making the (as far as I know) new proposal that dimer formation reactions may be accompanied by HNO3 loss, as a possible explanation for the C20 compounds observed with only a single N. The authors also describe observed thermal desorption profiles that hint at monomers coming from dimer dissociation in the instrument inlet, an important caution to other researchers using this technique. For these reasons, I think this manuscript will be a valuable contribution to the atmospheric chemistry literature, but it needs substantially more work before it is ready for publication. First, in many places I found the writing confusing and had a difficult time understanding what the authors were saying – I think this draft was at least one round of serious editting away from being ready to submit. I'll flag the sentences I found most confusing below, but after edits, I urge the authors to also find another outside reader to go through the entire manuscript. Second, and more importantly, the analysis of this rich dataset feels incomplete. At present, the paper really just presents FIGAERO MS and thermogram results, cluster analysis, and the HNO3 loss mechanism idea, without much support beyond the chemical formulae observed. I make the following broad suggestions for the authors to further this analysis before resubmitting for publication:

**Reply:** Actually, the manuscript had been sent for English language editing service. However, we might have missed that some of the changes suggested by the English language editing back-fired on the scientific clarity before submission. The paper has now been read again and we have addressed the issues raised by the referee.

(1) present a better foundation for the idea of HNO3 loss: show a proposed structural mechanism for how this would occur, cite additional literature on the enegetics of such a reaction and of any other experiments that made observations consistent with this proposal, and try to produce an (even coarse) nitrogen budget from your experiments to check if it supports HNO3 loss. How much of the N2O5 you injected didn't show up in gas or aerosol phase CIMS products? is any change in this budget over different experiments consistent with the amount of the products you hypothesize to come from this type of reaction?

**Reply/action:** There are some suggestions in the literature, however, it is not as clear as one hope it should be and we cannot firmly conclude this so we have to leave that open for speculations and following up-studies on simpler systems. The nitrogen budget would be useful but since $HNO_3$ is notorious to be "sticky" and we did not measure the other suspected "leaving groups" like $NO_2$ and NO we have poor handled on this. However, it is clear that the selected dimers only contain one nitrogen and have lower O/C ratio than the sum of two monomers. Furthermore, the Tmax suggests a high Mw compound (with low vapor pressure) and not a fragment. We now refer to some recent literature on organic nitrate condensed phase reactions.

(2) The cluster analysis should also be analyzed and discussed in greater detail: thus far, it seems to serve only to highlight the same two "groups" as the MS alone would have – the monomer region (in 3 factors) and the dimer region ( in 2 factors). You mention the families in each cluster and have generic chemical formulae for them. Can you propose structures / reasons these would be different?

Are they potentially from oxidation at the different double bonds (what would you expect that to look like?) or form RO2 + RO2 reactions vs. RO2 + NO3 reactions? Why is cluster 0 basically spread across the O;C and MW space of clusters 1 and 2 – why is it nevertheless a separate cluster? Is there any proposed mechanism that would get you this, or could it be that cluster 0 are the fragments of dimers while 1 and 2 are straight monomers? Or, some permutation of this?

**Reply:** Maybe, this was the information that was not so clear. However, it is actually stated that cluster 0 is not similar to the monomer cluster 1 and 2 but rather a fragmentation cluster having similar Mw and O/C but different Tmax (much higher). The reason for family m5 and m7 residing exclusively in factor 1 (higher volatility) is not known but clearly interesting results.

**Action:** The language has been polished further and checked.

(3) It seems you should also be able to do more with the fact that some monomers have double peaks and some don't – can you correlate this to the cluster analysis somehow, or otherwise interpret it mechanistically?

**Reply:** Yes, this is already included in the analysis. Some ions has two desorption maximum and then $T_{max,2}$ was also added in the analysis. E.g. cluster 0 contains mainly ions with a $T_{max,2}$

(4) You mention that some observed formulae are in MCM and others are not. Do more with this. Are there any structures predicted to be major products in MCM that you don't observe? I would suggest to show a modeled output of MCM (just a box model) for your expt. conditions. Are the major few predicted products all those that you observe, or are you just observing one product channel / some at random / etc.? You have the data now to truly test the MCM (and you allude to it), and I was disappointed to see that you didn't report a true comparison. For the major formulae that you observed that aren't in MCM, too, I'd like to see more analsysis. You say in your conclusions that these should be included in models, but you haven't told us what they might be. Propose some structures / mechanistic origins / something based on MCM to help guide how your novel observations might be incorporated.

**Reply:** As stated in the paper 69% of the gas-phase compounds could be referred to MCM compounds (assuming a common sensitivity).

**Action:** The MCM run provided a list of major compounds that is now presented in the supplemental information.

(5) Suggest to read this recent NO3 + limonene paper from the Ng group: http://pubs.acs.org/doi/abs/10.1021/acs.est.7b01460 and include comparisons to these results in your updated draft. Can you make any quantitative estimate of SOA yields to compare to Boyd et al.'s (very high!!) SOA yields from NO3 + limonene, or comment on them based on your mechanism? Are >100% yields consistent with your proposed dimerization mechanism, or inconsistent? Please discuss.

**Reply:** We realized that this is a recent study that was done in parallel to our work. However, if not necessary we prefer not to explicitly present aerosol yields from a flow reactor study. Furthermore,

with a very different method we abstain also from comments on their high yield .The dimerization mechanism would if anything enhance an aerosol yield since it will increase the observed Tmax.

**Specific editorial suggestions:**

in title: "nitrate-radical-initiated" should be "nitrate radical-initiated", and elsewhere in text

**Action:** This has been changed accordingly.

line 11 "reaction of NO3 with limonene"

**Action:** Text changed to recommended text (replace "and" to "with")

line 14 "identity and volatility of the most" Sentence line 17-19 is confusing. How about: "The observed products were compared to those in the Master Chemical Mechanism (MCM) limonene mechanism, and many non-listed species were identified."

**Action:** Text changed to read "Major condensed-phase species were compared to those in the Master Chemical Mechanism (MCM) limonene mechanism, and many non-listed species were identified. The volatility properties of the most prevalent organic nitrates in the produced SOA were determined."

line 39-40: "The oxidation of VOCs by"

**Action:** Text changed to read "The oxidation of VOCs by"

line 49: find some more recent refs to add to cite list

**Action:** reference list updated according to reviewer's suggestions

lines 55-58: seems odd not to have the NO3 + R formation of organonitrates listed here, since this is the one you focus on

**Action:** To be added

line 97 "molecular formulae of major nitrate species produced"

**Action:** Text changed to read "molecular formula of major nitrate species produced"

line 100 "(based on the Master Chemical. . ."

**Action:** Text changed to read "(based on the Master Chemical Mechanism)"

line 154 "1.4 g cm-3 (Hallquist. . ." { units}

**Action:** Units have been added

line 167: mention typical [NO3] here too, and comment on whether RO2 + RO2 or RO2 + NO3 reactions should be dominant? in caption for Table 1, mention hos you know the [N2O5]. Can you use these values to get an approximate SOA yield?

**Reply:** If not necessary we prefer to not explicitly present aerosol yields from this flow reactor study since the experimental design was not aimed at that purpose. The $N_2O_5$ was assumed to, under dry conditions, be equal to the amount added into the system. [NO3] was not measured.

line 210 "silhouette score, s(i) (Rousseuw. . .."

**Action:** Text changed to read "silhouette score, s(i) (Rousseuw

around line 236-240: "regions" is confusing – define as the designated mass ranges if you'll use for further discussion. 237-238 "These regions corresponded to aerosol samples" : what does this mean? They were always enhanced in aerosol samples relative to gas, or . . .?

**Action:** The regions are now better defined as low and high molecular weight regions and connected to monomer and dimer identification. The statement is basically refereeing to that elevated ion counts were always present in these region for all the aerosol measurements. This is now rephrased.

line 245: you refer to specific exptal condition of ratio = 2.4. This sounds contradictory to your statement that the regions "always" correspond to aerosol (I interpreted this as meaning under all ratios of reagents)

**Reply:** We selected ratio 2.4 as one example of all the ratios studied and not a specific exceptional condition. We don't claim it to be exceptional in the text? Rather we follow up showing the Figure 4 where there is a gradual change in monomer/dimer intensities as the $N_2O_5$/Limonene ratio changes

line 252: you call C11 a dimer? Also, the carbon number ranges here don't correspond to your shaded regions in Fig. 2, shouldn't they?

**Reply:** The definition was based on that C11 compounds must be produced from two carbon containing entities. The definition of monomer/dimers has in atmospheric science been used in a very reluctant way compared to traditional chemistry science, however, we prefer to stick to the more relaxed definition previously used. However, we agree that the carbon numbering does not fit with the figure. Basically, the values are calculated on HR-fit data selected based on carbon number stated while the shading in Figure for simplicity was based on m/z. We agree this is not consistent.

**Action:** This is now revised and clarified in the text.

line 257: now you're introducing a new designation, the "first" region for what you've called the "monomer" region before (I assume these mean the same thing). I urge you to define a term that refers to these regions at the beginning of this section and stick with it. Even better: label it on Fig. 2.
**Action:** First is changed to monomer

line 272-273: confused by "at least modestly" ?? wouldn't this just be the remaining 56.5%? I wouldn't call that modestly. Or I don't understand to what you refer here.

**Action:** Text changed to state significant contribution

line 274-275: confused by "the monomers contain": many? all? on average?

**Action:** This sentence is confusing and has been removed

line 276: "Progressively more oxygenated monomers of the general . . ."

**Action:** Text now reads "Monomers with Progressively more oxygenated monomers of the general formula $C_{10}H_{15}NO_x$"

line 277: omit the paranthetical

**Action:** Parentheses removed

line 283: how would you know a structure is a PAN? Unless you show a mechanism that would get to those species, I'd just leave this discussion as PNs only, they would all readily dissociate.

**Action:** Here we only state that they may be PNs or PAN like . The discussion on PAN like compounds are removed as suggested.

line 292: where did the range of formulae you cite for ELVOC come from? Is this your defition, or does it come from Donahue? (similar question for dimer formulae on line 310)

**Action:** Text amended to read "i.e., ELVOCs, which play a key role in SOA formation (Donahue et al., 2012)."

line 299: confused by "using α-pinene." using it as what? Nighttime SOA at SOAS is more likely to be from β-pinene, since its NO3 SOA yields are higher. Not sure what you mean here by invoking α-pinene.

**Action:** The Lee et al study used lab experiments on a-pinene to match observations in the field. The sentence has been revised to: "The gaseous parent compounds were identified as monoterpenes, matching ions measured in a laboratory study on α-pinene."

lines 299-303: rework into one sentence; repetitive; confusing. (Maybe because of trying to tie back to the a-pin reference? which you could just remove)

**Action:** removed as suggested by referee

line 304-305: not clear what this sentence means.

**Action:** removed as a consequence of the point above.

line 310-311: parenthetical seems internally contradictory: "only" in aerosol phase, but also "slightly about background" in the gas phase?

**Action:** The text has been amended to read "(i.e., they were present only in the aerosol phase and at insignificant levels in the gas samples)."

line 314 "ion families (defined as groups of molecular formulae with only the number of O atoms varying) to the total . . ." {correct?}

**Action:** The text has been amended to read "The contributions of the 11 most prevalent ion families (defined as groups of molecular compositions with only the number of O atoms varying) to the total desorbed organic signal are summarized in Table 2."

Table 2: the large ranges but with very precise end points for the Tmax reported is weird. If you really think only the range is interesting, truncate sig figs. If as I suspect the actual list of Tmax values for each member of the family (there are always 6 or fewer, so not too crazy) might be interested to aide your interpretation, why not list them instead, on the order of x value to people can see which correspond? then you can discuss more about which Tmax's appear across monomers / dimers and do some analysis / inference about connectedness from those!

**Action:** The Tmax values have been truncated as suggested by the referee.

line 328 "Additionally, (iii) some. . ."

**Action:** Text changed to read "Additionally, (iii) some"

line 346-348: how is this indicative of monomer contributions? do you mean indicative of LOWER monomere contributions to aerosol? Fig 3: why are some traces shaded to zero? describe/ discuss or don't have the difference. in caption, "N2O5 ratio" should be "N2O5 to limonene ratio" , right?

**Action:** Basically, the compounds with low Tmax were also found in the gas-phase and we then suspect them to not be fragments of larger "dimer compounds" but rather being the same compounds in the condensed phase that evaporates without fragmentation.The sentence has been changed to:

"In general, compounds evaporating at relatively low temperatures were also found in the gas phase, indicative of monomer that partitioning between gas and particle phase."

line s 362 / 365: do the "few" and the "ten species" refer to the same subset? This para is confusing.

**Action:** These two sentences have been revised and are now reading:

"Monomer, i.e., lower-mass, species (C ≤ 10) desorbing at high temperatures could be produced as fragments via thermal degradation of higher-MW species. Some of these ions are matching the chemical composition ($C_{10}H_{16}O_4$, $C_{10}H_{17}NO_5$, $C_{10}H_{17}NO_6$, and $C_7H_{10}O_4$) of primary products within the MCM, accounting for (on average) 69.0 ± 10.8% of the signal detected in the gas phase. Here some possibilities are plausible, one could be that they are produced as monomer but are important building blocks in the dimer formation, thus thermally decompose back to monomers during desorption."

line s 367-369: not sure how / why this suggests favored dimer formation – confusing

**Action:** Se point above

line 370: another new designation, "high-MW"! dimer/oligomer? be consistent in how you refer to the different groups

**Action:** Changed according to suggestions above.

line 373: cite Figure 4

**Action:** Text now refers to Figure 4

line 374: do you mean "the monomer signal at higher N2O5 to limonene ratio"? If so, say so. If not, clarify what you mean

**Action:** The sentence has been modified and now reads: "At high ratios of $N_2O_5$ to limonene, the fraction of dimer species decreased, whereas the percentage of monomer species (fragments) with high Tmax increased (Figure 4)."

line 375-376: "The average . . . Fig. 4" : remove this sentence, not clear & not helpful, and can cite Fig. 4 above instead.

**Action:** The sentence has been removed

line 407: define oxidation state here, where you first mention it, instead of later where you currently define it Fig. 5: put the labels for mass, desorption temperature, O/C and oxidation state in the same order (1) on the plot, (2) in the caption (and mention all 4!), and put the y axes in the same order, to make life easier on your readers. Also, you show S.D. on the plot (I guess? or, what are the error bars?) – mention this in caption.

**Action:** Oxidation state now defined in Line 400. The figure has been changed and the reference to SD is now given in the caption.

 do you mean "and are also represented as members of cluster 0"? i.e., the ions are members of both clusters? Clarify.

**Action:** The text now reads "These results from the fact that 87% and 69% of cluster 1 and 2 ions, respectively, have secondary thermogram peaks and $T_{max}$ values, and are represented as members of both clusters 1 and 0.

lines 424-428: Could an alternative rationale for this difference in oxidation state simply be that monomers, because of their small carbon chain length, need more oxidized functional groups to condense, while dimers are so big they'll condense even with less oxidation? Since you are looking only at the aerosol phase here, this pattern could be skewed by differing volatility.

**Action:** Yes, this could be one reason. This alternative rationale is now included in the discussion.

line 437: oxidation state is defined here, should be at first instance of it earlier in text.

**Action:** Oxidation state now defined in Line 400

line 438: @ "certain range of masses", state the range

**Action:** The wording has been changed so it is now clear that the dependence are within respectively group and not between the dimer and monomer groups. "A positive correlation between the average O/C ratios and $T_{max}$ values was obtained within monomer (clusters 1 and 2) and dimer (clusters 3 and 4) groupings, with more highly oxygenated species desorbing at higher temperatures than their less-oxygenated counterparts. This is also reflected in the overall oxidation state of the clusters. It should be noted that these correlations do not hold between the different groups since monomer species had (in general) higher O/C ratios than the dimers."

line 440: same comment as lines 424-428.

**Action:** ok, see comment on line 424-428

line 445 "during the process, and"

**Action:** Text changed to read "during the process, and"

line 446: as mentioned in general comments above, I think it would be good to include a structural diagram of the HNO3 leaving reaction.

**Action:** Even if the mechanism on $HNO_3$ as the leaving group (in a reverse esterification process) is plausible we now after the review comments would be a little bit more cautious. This cautiousness has now been implemented in the manuscript and in line with that we do not want to be explicit in this formation. However, we include that this could be seen as a reverse esterification but also might be linked to observed hydrolysis of organic nitrates (Rindelaub et al, 2016).

lines 450 & 452: since this relied on having C9 monomers, it would be useful also to explain how these are formed in the mechanism leading up to dimerization.

**Reply:** There are several pathways for C9 monomer compounds. E.g. if both double bonds are oxidizes and there is a C-C bond breakage for the exocyclic double bond it will create a C9 monomer. Additional oxygens could be produced by internal H abstractions.

line 453: at the end of this discussion, I think it would also be good to at least mention any alternative possible pathway that could make these dimer products – is it really only possible with

HNO3 loss, or could you get it some other way? Then, perhaps the nitrogen balance or other evidence can help you bolster your hypothesis that the HNO3 loss is the more likely route.

**Reply/Action:** Yes, there are some suggestions in the literature, however, it is not as clear as one hope it should be and we cannot firmly conclude this so we have to leave that open for speculations and following up-studies on simpler systems. The nitrogen budget would be useful but since $HNO_3$ is notorious to be "sticky" and we did not measure the other suspected "leaving groups" like $NO_2$ and NO we have poor handled on this. However, it is clear that the selected dimers only contain one nitrogen and have lower O/C ratio than the sum of the two monomers. Furthermore, the Tmax suggests a high Mw compound (with low vapor pressure) and not a fragment. We now refer to some recent literature on organic nitrate condensed phase reactions (e.g. Rindelaub et al, 2016)

line s 463-464: "dimer fragmentation . . . monomers." is a confusing sentence

**Action:** Fragmentation has been replaced and the sentence reworded. The sentence now reads:

"Due to the loss of $HNO_3$ during dimerization, the potential dimer decomposition during desorption is expected to yield fragments which differ in molecular composition from the precursor (i.e., pre-dimerization) monomers."

line s 465-469: can't you demonstrate this more conclusively by looking at specific examples of masses with and without double peaks, and possibly matching up the Tmax's, and thus identify the subset of monomers that are also dissociation products and possibly connect them to precursors? Even better – if relative amounts change with different reaction conditions, can you track them rising and falling together? This dataset would seem to have lots of potential to demonstrate these actually linkages, not just make vague statements and what might / could yield what else upon fragmentation.

**Reply/Action:** In theory, this could be a way forward. However, the complexity and numbers of free parameters will put constraints on such detailed analysis. Actually, the suggestion is very nice but should be done on simpler systems where more solid conclusions on this mechanism could be done. The suggested mechanism is now brought forward with cautiousness and this is now supported with statements on other possible leaving groups and reference to previous work.

line 480: "obtained for both gas and condensed-phase" – since you don't discuss gas phase data in this paper, omit, or cite to the companion paper that does study gas phase?

**Action:** Rephrased to state only particle phase "High-resolution mass spectrometric data was analysed for condensed-phase reaction products resulting from $NO_3$ initiated oxidation of the monoterpene, limonene."

line 484-485: lots of sig figs on these percentages considering the error bars – better to say 63 +/- 7 and 37 +/- 7 %? And perhaps put a few numbers in the abstract?

**Action:** Number of significant figures has been reduced

line 494-496: this has me wondering whether you can learn anything from the relative intensities of the two peaks? would this pattern support that SOA is "largely determined" by low-volatility oligomers?

**Reply:** If the two peaks are two different compounds with two different formation mechanisms and relative contribution to SOA there is little connection between them as for any pair of compounds detected. So we would not draw those conclusions from these observations.

line 504-505: as mentioned in the general comments, I think you should at least do some discussion of the non-listed products and what they might be.

**Action:** The products that were not found in the MCM list are now marked in the full list found in the supplemental. A comment on these and potential sources are now included.

line s 509-512: are all the hypothesized HNO3-loss dimers in one cluster and others in another? Or, how else can you use the cluster analysis to learn something related to your mechanism speculation?

**Action:** The cluster analysis clearly separated the fragments from the dimers (obviously, since a fragment cannot been formed according to the mechanism) and also classified the dimers in two groups.

line 513: "FIGAERO" to be consistent with how you write it above

**Action:** Text changed to read "FIGAERO-CIMS"

line 514: "may provide some means of reducing the complexity. . ." see above general comments. I hope you can work get a bit more out of this.

**Action:** The text is modified to capture extended use of cluster analysis and how this can be extrapolated and compared to other systems.

line 520-521: as mentioned above, hard to include species in modeling studiest if you haven't even suggested what you think they might be.

**Action:** The compounds are now listed in the supplemental and a short note on potential for extension of the MCM is given.

**Response to Anonymous Referee #3**

This work investigated organic nitrate formation from NO3 oxidation of limonene. Experiments were conducted using different N2O5/limonene ratios. Speciated gas and particle phase organic nitrates were measured by the FIGAERO-HR-ToF-CIMS. Cluster analysis of the desorption temperatures of organic nitrate species resulted in five clusters; the relationships between O/C, OS, MW, etc of these clusters were discussed. Formation of dimers was observed and reaction mechanisms for dimer formation were proposed.

This is an interesting study and the manuscript is generally well-written and easy to follow. This study will be of interest to the greater atmospheric community. My main comments are 1) while the experiments were conducted over a range of N2O5/limonene ratios, the authors shall provide more context to this experimental design. Also, the results from experiments with different N2O5/limonene ratios need to be more extensively and clearly discussed. 2) I have some concerns regarding the discussion of the results shown in Fig. 5, please see details below. 3) There are a number of recent studies on nitrate radical oxidation of biogenic hydrocarbons, it would be appropriate that these studies are referenced in the manuscript to reflect the current state of knowledge. Overall, I recommend publication in ACP once these comments are addressed. Most comments are mainly to improve clarity of the manuscript.

**Main Comments**

1. Page 5 line 158. Are potential impurities (e.g., NO2 and HNO3) in the N2O5 synthesized measured and quantified? Please make this clear in the manuscript.

**Action:** This is now clarified in the experimental: "$N_2O_5$ was synthesized by reacting ≥20 ppm $O_3$ with pure $NO_2$ (98%, AGA Gas) in a glass vessel and then passing the flow through a cold trap maintained at -78.5°C using dry ice. Even if neither $HNO_3$ nor $NO_2$ was measured it is known from previous work that this method typically provides a source with impurities less than a few percent. It is well known that the resulting white solid would show signs of yellowing, due to nitric or nitrous acid contamination, if exposed to moisture (e.g., ambient lab air) so handling of the $N_2O_5$ was done accordingly."

2. Page line 168. What is the reason for performing experiments with different N2O5/limonene ratios? Please provide more context here.

**Reply:** It would reflect a gradual increase in reaction of the exocyclic double bond. At a ratio of 1:1 we expect $NO_3$ to primarily react with the exocyclic double bond, representing the typical case producing primary products, while increasing the ratio will add more possibilities for reactions also with the endocyclic double bond, reflecting secondary atmospheric chemistry.

**Action:** Short discussion on this has been added in the manuscript

3. Page 6 Table 1. a. With these N2O5/limonene ratios, are all limonene (and both double bonds?) reacted away? Please clarify and change the "limonene" in the table to either "reacted limonene" or "initial limonene". b. What is the RO2 reaction regime in these experiments? RO2 + NO3? RO2 + RO2? c. Can SOA yields be quantitatively calculated from the values in the table? If the "limonene" is

reacted limonene, the SOA yields appear to be very low compared to previous studies by Fry et al. (ACP, 2009), Fry et al. (ACP, 2011), and Boyd et al. (ES&T, 2017). Please discuss the results from this study in the context of these prior studies. Also, do the data shown in Table 1 follow a typical Odum 2- product yield curve?

**Reply:** If not necessary we prefer not to explicitly present aerosol yields from this flow reactor study. The experiments were not designed to derive yields.

4. Page 8 line 238. It was noted that ". . ..the relative signal intensities varied with the amount of limonene and N2O5 present in the system". I think the authors are referring to Fig. 4? Please add the figure number to the sentence to help guide the readers if this is the case.

**Reply:** Yes, the referee is right but we then realize that this figure would then be introduced too early in the manuscript.

**Action:** We have removed the statement "and the relative signal intensities varied with the amount of limonene and $N_2O_5$ present in the system."

5. Page 8 line 260. It is not immediately clear what these species are without diving into the entire MCM mechanisms. The authors should at least include the formation mechanisms of these major ions in the SI to help guide the readers. Also, it would be helpful to propose mechanisms for the major species that were observed in this study but are not in MCM. On the related note, Boyd et al. (ES&T, 2017) recently expanded the limonene + NO3 mechanisms in MCM. It might be worthwhile to evaluate if some species detected in this study are covered in the expanded mechanism in Boyd et al.

**Reply:** The Boyd et al. paper was unfortunately not available when the majority of the analysis in the present study was done. Some of the species in the expanded mechanism have now been considered.

6. Page 9 line 280. What are some of the mechanisms for limonene and its oxidation products to react with NO2 and HNO3?

**Reply:** The possibilities are for example that $NO_2$ could react with peroxy-radicals and $HNO_3$ might do condensed phase nitration. We don't think that this of general value and would not add that.

7. Page 10 line 293. Nah et al. (ES&T 2016) also measured a large suite of highly oxygenated organic nitrates from NO3 oxidation of a-pinene and b-pinene in laboratory experiments, using the FIGAERO-HR-ToF-CIMS. Many of those are also observed in Lee et al. (PNAS, 2016).

**Reply:** We have noted the request for including Nah et al. (ES&T 2016).

**Action:** As the last sentence in this paragraph we added: Recently, Nah et al. (ES&T 2016) also measured a large suite of highly oxygenated organic nitrates from NO3 oxidation of a-pinene and b-pinene in laboratory experiments.

8. Page 10 line 299. It is noted that "The similarity with ions from the NO3-initiated limonene oxidation further emphasizes the importance of monoterpenes as precursors of organic nitrates". It

would be informative if the authors indicate in the Table in SI (the ion list) regarding which ions have also been observed in the ambient (Lee et al.) and other monoterpene experiments (Nah et al).

**Reply:** We would prefer to focus on comparison with limonene studies. It turns out there is now a parallel study by Boyl et al, 2017 (accepted when this manuscript was submitted) that would be better suited for comparisons.

9. Page 10 line 301. Rollins et al. (Science, 2012) discussed the importance of limonene + NO3 in a field study.

**Action:** This reference has now been added

10. Page 12 line 362. It was noted that there is a positive correlation (R2 = 0.67) between Tmax and molecular mass. Are the authors referring to Fig 5a? If so, it does not look like the overall correlation is that good? Please clarify.

**Reply/action:** There were no figures showing this correlation. However, we have now added one figure in the supplemental showing this correlation.

11. Page 13 line 370-379. (this is also related to comment #2 above). It is not clear to me how the higher N2O5/limonene experiments lead to formation of more thermally unstable mechanisms. Please explain. More discussions are needed here, to provide context to why experiments are conducted with different N2O5/limonene in the first place and why/how the resulting compositions are different.

**Reply:** This is discussed in the above #2 comment

12. Page 13 Figure 4. What is the function used for the fit? Is this just to guide the eye or there is a fundamental reason for such a dependence?

**Reply / Action:** This fit has now been removed, as discussed in a previous comment it illustrated a general best fit for the eye of the reader

13. Page 15, discussions of Figure 5. The authors attempted to discuss the relationships between O/C, OS, and MW, etc. However, within uncertainties, there do not seem to be significant differences in the O/C and OS values for all clusters. Hence, this discussion needs to be revised.

a. (related to comments # 2 and 11 above). In Figure 5, will any specific patterns emerge if the authors only look at the data from experiment of a particular N2O5/limonene ratio?

b. line 418. It was noted that the O/C of cluster 0 is similar to clusters 1 and 2. However, within the uncertainties, the O/C ratios of all clusters are almost the same.

c. line 426, should the high-MW clusters be (3,4)? And the low-MW clusters be (0, 1, 2)?

d. line 427. It was noted that the ions in the high-MW clusters have a lower OS than ions in clusters 0-3. Firstly, should "0-3" be "0-2"? Secondly, it does not look like the high-MW clusters have a lower OS. Within uncertainties, the OS values appear to be the same for all clusters.

e. line 434. It was noted that a positive correlation exists between O/C and Tmax. It is not clear how this is case from the data shown in Fig. 5c. Please provide a figure and show the R2 value.

**Action:** This section has now been revised considering the points addressed by the referee.

**Minor Comments**

1. Page 2 line 44. Would be appropriate to reference Ng et al. (ACP, 2017).

**Action:** Reference has been added

2. Page 2 line 46. Would be appropriate to also reference Day et al. (AE, 2010); Fry et al. (ACP, 2013); Xu et al. (PNAS, 2015); Xu et al. (ACP, 2015); Boyd et al. (ACP, 2015); Kiendler-Scharr et al. (GRL, 2016); Nah et al. (ES&T, 2016).

**Action:** References have been added

3. Page 2 line 60. Delete "M" in front of Hallquist.

**Action:** The "M." has been removed

4. Page 3 line 77. Boyd et al. (ES&T, 2017) recently investigated SOA formation from NO3 oxidation of limonene.

**Action:** Reference has been added

6. Page 14 Figure 5. Missing y-axis label for 5b?

**Action:** This is fixed now.

---

## Author Response (AR1)

We thank the reviewers for the helpful comments! We have now taken the benefit from those in improving the manuscript. A point by point response by a reply or/and actions (in black) to the reviewers' comments (in blue) will follow. New texts added or removed are shown in italics. The aim was to reply on each comment separately even if it creates some repetitions in replies (several comments solved by similar change). However, due to the number of comments there are a few that become obsolete due to changes from others. Then that are described by a reply. At the end is the full manuscript with changes in "track changes mode".

**Response to Anonymous Referee #1**

This manuscript describes data collected from a FIGAERO-CIMS during laboratory studies using the flow reactor GFROST. Focus is given on the major organic nitrate constituents of secondary organic aerosol from the NO3 oxidation of limonene. A cluster-analysis method is used in order to distinguish the groups of ions based on their thermodynamic properties and molecular formula information. Mechanisms of dimer formation are suggested based on the grouping performed. The paper is suitable for publication in ACP. My suggestions below are mainly to clarify the presentation

**Reply:** Thanks for the suggestions to clarify the presentation.

Specific comments 1. Recent studies have shown that fragmentation due to thermal dissociation in the FIGAERO-CIMS could strongly affect the chemical formula attribution (Stark et al., 2017). It would be beneficial if a comparison of the different methods of assigning a chemical formula to the major ions could be performed. If not, then the possible uncertainties introduced should be further discussed.

**Reply:** Yes, we should more clearly acknowledge the possibilities for decomposition but would rather be brief and refer to the findings in the Stark et al paper.

**Action:** Reference added with short notice on fragmentation.

*"Recently, Stark et al. (2017) showed that fragmentation during the desorption can occur within the FIGAERO. In the current work the fragmentation within the FIGARO was not specifically investigated. However, from our cluster analysis it was evidently that fragmentation occurred with specific features in e.g. molecular weight and evaporation temperature. The ramp rate during desorption was therefore maintained for all experiments to ensure, if fragmentation did occur, it would be consistent and enable comparable analysis of the dataset"*

2. In line 98 it is mentioned that one of the main focuses of this paper is to determine the molecular formula of species that could contribute significantly to SOA formation. Although the uncertainty of calculating the mass concentration using FIGAERO-CIMS would be high it could still indicate whether the compounds measured are indeed a large fraction of the overall mass as has been done from Isaacman-VanWertz et al. (2017).

**Reply:** Assuming common sensitivities for all compounds would enable the reader to find the major contributors to SOA mass in the list of compounds (Table S1 supplemental). The application of various sensitivities for I-CIMS is to date very uncertain and discussed in the literature, e.g. Isaacman-VanWertz et al. (2017).

**Action:** The Isaacman-VanWertz et al. (2017) has been added as reference for current state where they summarising methods to derive concentrations from I-CIMS . In addition, a short notice on the issue brought up by the referee with reference to Table 1 has been added.

*"This list is based on a common sensitivity for detection that might not always be true and highly variable (see e.g. Isaacman-Van Wertz et al, 2017). However, with this assumption the list will provide molecular identity of the most prominent organic compounds contributing to the SOA mass outlined in Table 1. One could assess the contribution of these peaks to the total mass loading, although with high variation in molecular mass and oxidation, the sensitivity is likely to vary significantly, resulting in large error margins and therefore deeming any interpretation highly speculative."*

3. FIGAERO-CIMS collects both the gas and particle phase compounds on the filter. More information on the gas phase compounds detected would bring light concerning the extent of possible gas phase "interference" during desorption. When the signal of the compounds in the gas phase is high then their contribution to the aerosol during desorption could increase more. Have there been any checks on the contribution of gas phase signal during desorption? How much is it expected to be in the monomer and how much in the dimer range? A figure to illustrate this effect in the supplementary would be very informative. For example, in lines 346-348 this could also be due to gas phase compounds collected on the filter that undergo evaporation and thus contribute to the aerosol. What is the gas phase concentration of these compounds compared to the aerosol?

**Reply**: The methods and potential artifacts have been described in previous publications. Generally, there are limiting gas-phase contaminations of the condensed phase evaporation. Yes, the gas-phase has also been measured and could be presented to illustrate the ratio between gas and particle phase concentrations.

**Action:** Two figures illustrating partition (ratio between the particle and the gas phase) for ions in monomer and dimer region has been added in supplemental (Fig S1). A note at the end of first paragraph in section 3.1 now reads:

*"The gas to particle ratio of most ions were below one as illustrated in Fig. S1, whereas the focus of this work was to characterize the particle phase."*

4. In Figure 3 for many of the compounds there is a residual signal above zero. How would that affect the signal of the next desorption? When FIGAERO reaches stable conditions what does that exactly mean? It would be fruitful for the reader if background measurements could be provided during these experiments on the Supplementary material. An additional figure to support the stability of the thermograms would be very instructive. For example the average cumulative signal (with error bars indicating the standard deviation of the average) vs temperature, for each compound, with each compound indicated with a different color would be a suggested way.

**Reply:** Usually several cycles are repeated and an average of three desorptions are used for the results

**Action:** Examples of three consecutive desorptions are now presented in supplemental (see Fig S2). The text refers to this figure in the experimental part. It now reads:

*"An average of four sequential desorption with corresponding standard deviation is shown in Fig. S2.*

5. What is the atmospheric relevance with the mixing ratio of N2O5 used? What is the NO3 expected mixing ratio in the system? A discussion would better inform the reader.

**Reply:** Obviously, the mixing ratio of $N_2O_5$ is higher than ambient to provide enough amount of limonene reacting during the time spent in the flow reactor. At a ratio of 1:1 we expect $NO_3$ to primarily react with the exocyclic double bond, representing the typical case producing primary products, while increasing the ratio will add more possibilities for reactions also to the endocyclic double bond reflecting secondary atmospheric chemistry.

**Action:** Short discussion on our view on this has been added in the in experimental:

*"At a ratio around 1.0 one expects only the endocyclic double bond to be reacting with $NO_3$ radicals while at higher ratio there is an increased possibility for secondary chemistry where products will be susceptible for reaction with the $NO_3$ radical."*

6. In Figure 4 what is the fit function used? What is the information we gain from this fit choice? Further discussion in the manuscript would clarify these questions. This figure doesn't provide all the data points seen in Table 1. The high N2O5/limonene ratio of 113.3 is missing. Could that be included or provided in the supplementary? Is this point included in the fit function applied now? Finally, error bars for the y-axis are missing and the fit function should be applied by taking into account this error.

**Reply:** yes, we agree the line is mostly for guidance of the eye and the r2 should be removed. The data point at 113.3 was not illustrated as it would extend the x-axes too far.

**Action:** The r2 has been removed and the missing data point at 113 and its corresponding values are now noted in the footnote. The end of the Figure caption now reads:

*"The data points at a ratio of 113 are not shown (22, 78, 39%, respectively). The lines indicated are for the guidance of the eye."*

7. In section 2.3 was it both gas and particle phase data used as identified species to compare to the MCM? A sentence to make clear that this work is focused on the particulate phase and not the gas phase would better direct the reader at this point.

**Reply:** The HR fitting procedure used ions both from gas and particle phase.

**Action:** A note has been added to direct the reader on this point at the end of the sentence:

"….the corresponding theoretical product distribution was compared with the measured distribution *for both gas and particle phase."*

To further stress the focus on condensed phase a sentence was added in first part of R&D.

*"The focus in the current work was on condensed phase products using the FIGARO inlet desorption"*

8. It would be beneficial for the reader if the discussion in section 3.4 was extended on how the suggested mechanisms can be directly inferred from the grouping technique and the volatility of the compounds.

**Reply:** The lower O/C ratios and loss of one nitrogen compound for some of the dimers while exhibiting a low volatility would support the mechanism suggested. (loss of oxygen(s) and nitrogen in dimerisation)

**Action:** This info has now been rewritten and the initial text now reads:

*"The mechanism to create dimers with one nitrogen and a lower O/C ratio would presumably involve the loss of a nitrogen oxides or nitric acid. For this complex system and within the scope of this study it was not possible to firmly proof any mechanism. Since the experiment were done at low RH the direct hydrolysis would be less likely (see Rindelaub et al, 2015, 2016). However, knowing $HNO_3$ being thermodynamic stable one may speculate in that dimerization of two monomer species via the loss of one $HNO_3$ molecule could occur e.g. where a $C_{20}H_{29}NO_y$ (y = 10–18) species would be generated from $C_{10}H_{15}NO_x$ (x = 5–9) species. This process could be seen as the reverse of esterification in order to produce a dimer product with one less nitrogen and reduced numbers of oxygens."*

**Technical comments**

Line 47. A broader view of organic nitrates is given by Kiendler-Scharr (2016).

**Action:** Reference added

Line 154. Units are missing.

**Action:** Units of $g\ cm^{-3}$ have been added

Line 167. Delete one dot.

**Action:** Full stop deleted

Line 234. Recent findings suggest that fragmentation due to thermal dissociation occurs in systems like the FIGAERO-CIMS (Stark et al., 2017). See comment 1.

**Action:** The following text has been added to acknowledge the work of Stark *et al.* (2017) and we have considered for possibilities that ions may fragment upon desorption:

*"Recently, Stark et al. (2017) showed that fragmentation during the desorption can occur within the FIGAERO. In the current work the fragmentation within the FIGARO was not specifically investigated. However, from our cluster analysis it was evidently that fragmentation occurred with specific features in e.g. molecular weight and evaporation temperature. The ramp rate during desorption was therefore maintained for all experiments to ensure, if fragmentation did occur, it would be consistent and enable comparable analysis of the dataset"*

Line 324-337. The authors provide four characteristic desorption patterns and the numbering stops at two.

**Action:** Addition numbering has been added in the text.

Line 361-362. It would be more informative to add the positive correlation in the supplement as a figure.

**Action:** The text refers to Fig. 5a. However, it has now been moved and clarified (since the trend is only valid for part of the data and Fig5a is discussed later in the text). It now reads:

*"A positive trend between the MW and $T_{max}$ values, see Fig 5a., was obtained for data in two of the monomer clusters (1 and 2) and the high volatile dimer cluster, while the trend turned negative for the low volatile dimers cluster."*

Line 387. Please define what extremely low signal would be as a value.

**Action:** A definition has now been added.

*"extremely low signal (i.e. the thermogram did not exhibit any structure identifiable above background noise prohibiting Tofware to constrain a mathematical fit for Tmax calculations)"*

Line 428. Should be 0-2?

**Action:** Changed to 0-2 in text

Table 1. Sorting of the lists would make the table more readable. Sorting the N2O5 from low to high and within each N2O5 group sorting the limonene from low to high would be one way of performing the sorting. Adding the expected NO3 concentration would be fruitful. Errors for the average SOA mass measured from an SMPS are missing. An additional column of the mass FIGAERO-CIMS could detect (with its much higher uncertainty), as discussed above, would be an indicator of how much of the overall SOA mass is measured in this system.

**Action:** The Table has been ordered as suggested by the reviewer. We did not measure the actual $NO_3$ concentrations and the total mass from FIGAERO-CIMS is rather uncertain. However, the standard deviation of the mean SOA mass is now included in the Table.

Table 2. What is the information we gain from the (N/C)x10? Based on all uncertainties the temperature precision could be rounded. Error of the average contribution is missing.

**Action:** The column displaying N/Cx10 has been removed. The temperatures have been rounded to the nearest degree. Errors of the average contributions are now included.

Figure 3. Certain double peak compounds are not highlighted like C10H15NO7, C20H29NO5, C20H24N2O8 etc. The C20H24N2O8 is not written correctly in the annotation. Since there are more compounds that have double peaks plotted it would be clearer to include the double peak compounds with dash lines and avoid highlighting.

**Action:** The highlighting has been removed and the mis-formatted compound name ($C_{20}H_{24}N_2O_8$) has been corrected.

Figure 5. It improves the figure if (a) and (b) have the same annotation introduced on the right side outside both figures once. This way you avoid the change in font size that is seen for annotation from Figure (a). For Figure (b) the oxidation state is not mentioned in the axis and the range is not going to minus when it is below zero. It would be beneficial if Figure (c) was separated in two graphs. On the left side a figure of the MW (left-axis) and Tmax (right-axis) and on the right side a figure of the oxidation state (left-axis) and the O/C (right-axis). Box-and-whiskers instead of bars and markers would provide more information on the dataset. This would also show more clearly the temperature increase that is suggested to correlate to the O/C increase. Finally the colors of Figure (c) are similar to the colors of the clusters thus confusing the reader.

**Action:** Changed the plot to a 4-panel plot by separating the bottom graph. The other minor formatting issues were also fixed, like the single legend for the top plots.

References Isaacman-VanWertz, G., P. Massoli, R. E. O'Brien, J. B. Nowak, M. R. Canagaratna, J. T. Jayne, D. R. Worsnop, L. Su, D. A. Knopf, P. K. Misztal, C. Arata, A. H. Goldstein, and J. H. Kroll: Using advanced mass spectrometry techniques to fully characterize atmospheric organic carbon: current capabilities and remaining gaps, Faraday Discussions, doi:10.1039/C7FD00021A, 2017.

Kiendler-Scharr, A.: Ubiquity of organic nitrates from nighttime chemistry in the European submicron aerosol, Geophys. Res. Lett., 43, 7735–7744, doi:10.1002/, 2016.

Stark, H., R. L. N. Yatavelli, S. L. Thompson, H. Kang, J. E. Krechmer, J. R. Kimmel, B. B. Palm, W. Hu, P. L. Hayes, D. A. Day, P. Campuzano-Jost, M. R. Canagaratna, J. T. Jayne, D. R. Worsnop, and J. L. Jimenez: c, Environ Sci Technol, doi:10.1021/acs.est.7b00160, 2017.

**Response to Anonymous Referee #2**

General comments: This paper presents novel flow tube measurements of the chemical composition of SOA produced from NO3 + limonene, with the potential to provide valuable new mechanistic clues and making the (as far as I know) new proposal that dimer formation reactions may be accompanied by HNO3 loss, as a possible explanation for the C20 compounds observed with only a single N. The authors also describe observed thermal desorption profiles that hint at monomers coming from dimer dissociation in the instrument inlet, an important caution to other researchers using this technique. For these reasons, I think this manuscript will be a valuable contribution to the atmospheric chemistry literature, but it needs substantially more work before it is ready for publication. First, in many places I found the writing confusing and had a difficult time understanding what the authors were saying – I think this draft was at least one round of serious editting away from being ready to submit. I'll flag the sentences I found most confusing below, but after edits, I urge the authors to also find another outside reader to go through the entire manuscript. Second, and more importantly, the analysis of this rich dataset feels incomplete. At present, the paper really just presents FIGAERO MS and thermogram results, cluster analysis, and the HNO3 loss mechanism idea, without much support beyond the chemical formulae observed. I make the following broad suggestions for the authors to further this analysis before resubmitting for publication:

**Reply:** Actually, the manuscript had been sent for English language editing service. However, we might have missed that some of the changes suggested by the English language editing back-fired on the scientific clarity before submission. The paper has now been read again and we have addressed the issues raised by the three referees which make the science more clear.

(1) present a better foundation for the idea of HNO3 loss: show a proposed structural mechanism for how this would occur, cite additional literature on the enegetics of such a reaction and of any other experiments that made observations consistent with this proposal, and try to produce an (even coarse) nitrogen budget from your experiments to check if it supports HNO3 loss. How much of the N2O5 you injected didn't show up in gas or aerosol phase CIMS products? is any change in this budget over different experiments consistent with the amount of the products you hypothesize to come from this type of reaction?

**Reply:** There are some suggestions in the literature, however, it is not as clear as one hope it should be and we cannot firmly conclude this so we have to leave that open for speculations and following up-studies on simpler systems. The nitrogen budget would be useful but since $HNO_3$ is notorious to be "sticky" and we did not measure the other suspected "leaving groups" like $NO_2$ and NO we have poor handled on this. However, it is clear that the selected dimers only contain one nitrogen and have lower O/C ratio than the sum of two monomers. Furthermore, the Tmax suggests a high Mw compound (with low vapor pressure) and not a fragment.

**Action:** We now refer to some recent literature on organic nitrate condensed phase reactions (hydrolysis) and clarify our speculation further.

*"The mechanism to create dimers with one nitrogen and a lower O/C ratio would presumably involve the loss of a nitrogen oxides or nitric acid. For this complex system and within the scope of this study it was not possible to firmly proof any mechanism. Since the experiment were done at low RH the direct hydrolysis would be less likely (see Rindelaub et al, 2015, 2016). However, knowing $HNO_3$ being*

*thermodynamic stable one may speculate in that dimerization of two monomer species via the loss of one $HNO_3$ molecule could occur e.g. where a $C_{20}H_{29}NO_y$ (y = 10–18) species would be generated from $C_{10}H_{15}NO_x$ (x = 5–9) species. This process could be seen as the reverse of esterification in order to produce a dimer product with one less nitrogen and reduced numbers of oxygens."*

(2) The cluster analysis should also be analyzed and discussed in greater detail: thus far, it seems to serve only to highlight the same two "groups" as the MS alone would have – the monomer region (in 3 factors) and the dimer region ( in 2 factors). You mention the families in each cluster and have generic chemical formulae for them. Can you propose structures / reasons these would be different? Are they potentially from oxidation at the different double bonds (what would you expect that to look like?) or form RO2 + RO2 reactions vs. RO2 + NO3 reactions? Why is cluster 0 basically spread across the O;C and MW space of clusters 1 and 2 – why is it nevertheless a separate cluster? Is there any proposed mechanism that would get you this, or could it be that cluster 0 are the fragments of dimers while 1 and 2 are straight monomers? Or, some permutation of this?

**Reply:** Maybe, this was the information that was not so clear. However, it is actually stated that cluster 0 is not similar to the monomer cluster 1 and 2 but rather a fragmentation cluster having similar Mw and O/C but different Tmax (much higher). The reason for family m5 and m7 residing exclusively in factor 1 (higher volatility) is not known but clearly interesting results. Generally, its obviously good that the cluster analysis confirm the more commonly "by the eye" definition on monomer and dimer regions from inspection of the mass spectra alone.

**Action:** The discussions and the actual Fig. 5 has been changed according to a number of other comments.

(3) It seems you should also be able to do more with the fact that some monomers have double peaks and some don't – can you correlate this to the cluster analysis somehow, or otherwise interpret it mechanistically?

**Reply:** Yes, this is already included in the analysis. Some ions has two desorption maximum and then $T_{max,2}$ was also added in the analysis. E.g. cluster 0 contains mainly ions with a $T_{max,2}$

(4) You mention that some observed formulae are in MCM and others are not. Do more with this. Are there any structures predicted to be major products in MCM that you don't observe? I would suggest to show a modeled output of MCM (just a box model) for your expt. conditions. Are the major few predicted products all those that you observe, or are you just observing one product channel / some at random / etc.? You have the data now to truly test the MCM (and you allude to it), and I was disappointed to see that you didn't report a true comparison. For the major formulae that you observed that aren't in MCM, too, I'd like to see more analsysis. You say in your conclusions that these should be included in models, but you haven't told us what they might be. Propose some structures / mechanistic origins / something based on MCM to help guide how your novel observations might be incorporated.

**Reply:** As stated in the paper 69% of the gas-phase compounds could be referred to MCM compounds (assuming a common sensitivity). Yes, we should have presented an output from MCM model to compare with observed condensed phase products.

**Action:** The MCM run provided a list of major compounds that is now presented in the supplemental information (Table S1).

(5) Suggest to read this recent NO3 + limonene paper from the Ng group: http://pubs.acs.org/doi/abs/10.1021/acs.est.7b01460 and include comparisons to these results in your updated draft. Can you make any quantitative estimate of SOA yields to compare to Boyd et al.'s (very high!!) SOA yields from NO3 + limonene, or comment on them based on your mechanism? Are >100% yields consistent with your proposed dimerization mechanism, or inconsistent? Please discuss.

**Reply:** We realized that this is a recent study that was done in parallel to our work. However, if not necessary we prefer not to explicitly present aerosol yields from a flow reactor study. Furthermore, with a very different method we abstain also from comments on their high yield .The dimerization mechanism would if anything enhance an aerosol yield since it will increase the observed Tmax.

**Specific editorial suggestions:**

 in title: "nitrate-radical-initiated" should be "nitrate radical-initiated", and elsewhere in text

**Action:** This has been changed accordingly.

line 11 "reaction of NO3 with limonene"

**Action:** Text changed to recommended text (replace "and" to "with")

line 14 "identity and volatility of the most" Sentence line 17-19 is confusing. How about: "The observed products were compared to those in the Master Chemical Mechanism (MCM) limonene mechanism, and many non-listed species were identified."

**Action:** Text changed to read:

*"Major condensed-phase species were compared to those in the Master Chemical Mechanism (MCM) limonene mechanism, and many non-listed species were identified. The volatility properties of the most prevalent organic nitrates in the produced SOA were determined."*

line 39-40: "The oxidation of VOCs by"

**Action:** Text changed accordingly line 49: find some more recent refs to add to cite list

**Action:** reference list updated according to reviewer's suggestions lines 55-58: seems odd not to have the NO3 + R formation of organonitrates listed here, since this is the one you focus on

**Action/reply:** This reaction is described in text but not in the specific reactions outlined as correctly pointed out by reviewer. However, since all reactions are described in text the reaction mechanism are obsolete and more "text book material". We decided its not necessary to present R1 and R2.

**Action:** Text *"Reactions 1–2 show the typical formation pathway of organic nitrates and peroxy nitrates formed from the reactions of NOx with RO2• originating from a generic VOC, R"* and the subsequent reactions has been removed.

line 97 "molecular formulae of major nitrate species produced"

**Action:** Text changed to read "molecular formulae of major nitrate species produced"

line 100 "(based on the Master Chemical. . ."

**Action:** Text changed to read "(based on the Master Chemical Mechanism)"

line 154 "1.4 g cm-3 (Hallquist. . ." { units}

**Action:** Units have been added line 167: mention typical [NO3] here too, and comment on whether RO2 + RO2 or RO2 + NO3 reactions should be dominant? in caption for Table 1, mention hos you know the [N2O5]. Can you use these values to get an approximate SOA yield?

**Reply:** If not necessary we prefer to not explicitly present aerosol yields from this flow reactor study since the experimental design was not aimed at that purpose. The amount of $N_2O_5$ was assumed to, under dry conditions, be equal to the amount added into the system. [NO3] was not measured.

line 210 "silhouette score, s(i) (Rousseuw. . .."

**Action:** Text changed accordingly around line 236-240: "regions" is confusing – define as the designated mass ranges if you'll use for further discussion. 237-238 "These regions corresponded to aerosol samples" : what does this mean? They were always enhanced in aerosol samples relative to gas, or . . .?

**Action:** The regions are now better defined as low and high molecular weight regions and connected to monomer and dimer identification. The statement is basically refereeing to that elevated ion counts were always present in these region for all the aerosol measurements. This is now rephrased:

*"These regions were present in all experiments (Table 1). The occurrence of ions in these regions indicates a prevalence of lower-mass monomer species (typically in the range m/z 340-440) and higher-mass dimer species (typically in the range m/z 580-700)."*

line 245: you refer to specific exptal condition of ratio = 2.4. This sounds contradictory to your statement that the regions "always" correspond to aerosol (I interpreted this as meaning under all ratios of reagents)

**Reply:** We selected ratio 2.4 as one example of all the ratios studied and not a specific exceptional condition. We don't claim it to be exceptional in the text? Rather we follow up showing the Figure 4 where there is a gradual change in monomer/dimer intensities as the $N_2O_5$/Limonene ratio changes. **Action:** More clear now after rephrase according to the previous comment.

line 252: you call C11 a dimer? Also, the carbon number ranges here don't correspond to your shaded regions in Fig. 2, shouldn't they?

**Reply:** The definition was based on that C11 compounds must be produced from two carbon containing entities. The definition of monomer/dimers has in atmospheric science been used in a very reluctant way compared to traditional chemistry science, however, we prefer to stick to the more relaxed definition previously used. However, we agree that the carbon numbering does not fit with the figure. Basically, the values are calculated on HR-fit data selected based on carbon number stated while the shading in Figure for simplicity was based on m/z. We agree this is not consistent.

**Action:** This is now revised and clarified in the text.:

*"From the HR analysis the definition of monomer and dimer was specifically defined based on number of carbons rather than the less strict used of the two m/z regions illustrated in Fig. 2."*

line 257: now you're introducing a new designation, the "first" region for what you've called the "monomer" region before (I assume these mean the same thing). I urge you to define a term that refers to these regions at the beginning of this section and stick with it. Even better: label it on Fig. 2.
**Action:** Rephrased to:

*"The lower-mass region, of the two mass-spectra regions (see Fig. 2) typically occurred at m/z values ranging from 340 to 440 and containing mainly monomers."*

line 272-273: confused by "at least modestly" ?? wouldn't this just be the remaining 56.5%? I wouldn't call that modestly. Or I don't understand to what you refer here.

**Action:** Text changed to "contributed significantly"

line 274-275: confused by "the monomers contain": many? all? on average?

**Action:** This sentence is confusing and has been replaced by:

*"One common feature of the monomers without a match in MCM is that they contain a nitrogen atom and have an oxygen number higher than 6, which is a range of compounds that is not represented explicitly in the MCM."*

line 276: "Progressively more oxygenated monomers of the general . . ."

**Action:** Text now reads "Monomers with progressively more oxygenated monomers of the general formula $C_{10}H_{15}NO_x$"

line 277: omit the paranthetical

**Action:** Parentheses removed line 283: how would you know a structure is a PAN? Unless you show a mechanism that would get to those species, I'd just leave this discussion as PNs only, they would all readily dissociate.

**Action:** Yes, we agree and have removed the discussion on PAN like compounds as suggested.

line 292: where did the range of formulae you cite for ELVOC come from? Is this your defition, or does it come from Donahue? (similar question for dimer formulae on line 310)

**Action:** Text amended to read "i.e., ELVOCs, which play a key role in SOA formation (Donahue et al., 2012)." At line 310 the formula was removed.

line 299: confused by "using α-pinene." using it as what? Nighttime SOA at SOAS is more likely to be from β-pinene, since its NO3 SOA yields are higher. Not sure what you mean here by invoking α-pinene.

**Action:** The Lee et al study used lab experiments on a-pinene to match observations in the field. The sentence has been revised to:

*"The gaseous parent compounds were identified as monoterpenes, matching ions measured in their laboratory study on α-pinene"*

lines 299-303: rework into one sentence; repetitive; confusing. (Maybe because of trying to tie back to the a-pin reference? which you could just remove)

**Action:** Shortened and replaced by

*", enforcing the importance of monoterpene nitrates in the ambient atmosphere."*

line 304-305: not clear what this sentence means.

**Action:** clarified and now reads:

*"For all elevated ion signals above m/z 390, there was no corresponding product in the MCM mechanism."*

line 310-311: parenthetical seems internally contradictory: "only" in aerosol phase, but also "slightly about background" in the gas phase?

**Action:** The text has been amended to read

*"(i.e., they were present only in the aerosol phase and at insignificant levels in the gas samples)."*

line 314 "ion families (defined as groups of molecular formulae with only the number of O atoms varying) to the total . . ." {correct?}

**Action:** The text has been amended to read

*"The contributions of the 11 most prevalent ion families (defined as groups of molecular compositions with only the number of O atoms varying) to the total desorbed organic signal are summarized in Table 2."*

Table 2: the large ranges but with very precise end points for the Tmax reported is weird. If you really think only the range is interesting, truncate sig figs. If as I suspect the actual list of Tmax values for each member of the family (there are always 6 or fewer, so not too crazy) might be interested to aide your interpretation, why not list them instead, on the order of x value to people can see which correspond? then you can discuss more about which Tmax's appear across monomers / dimers and do some analysis / inference about connectedness from those!

**Action:** The Tmax values have been truncated as suggested by the referee.

line 328 "Additionally, (iii) some. . ."

**Action:** Text changed accordingly.

line 346-348: how is this indicative of monomer contributions? do you mean indicative of LOWER monomere contributions to aerosol? Fig 3: why are some traces shaded to zero? describe/ discuss or don't have the difference. in caption, "N2O5 ratio" should be "N2O5 to limonene ratio" , right?

**Action:** Basically, the compounds with low Tmax were also found in the gas-phase and we then suspect them to not be fragments of larger "dimer compounds" but rather being the same compounds in the condensed phase that evaporates without fragmentation. The sentence has been changed to:

*"In general, compounds evaporating at relatively low temperatures were also found in the gas phase, indicative of monomer that partitioning between gas and particle phase."*

line s 362 / 365: do the "few" and the "ten species" refer to the same subset? This para is confusing.

**Action:** The paragraph has been revised and the part previously starting with "A few monomers." now reads:

*"Monomer, i.e., lower-mass, species (C ≤ 10) desorbing at high temperatures could be produced as fragments via thermal degradation of higher-MW species. Some of these ions are matching the chemical composition ($C_{10}H_{16}O_4$, $C_{10}H_{17}NO_5$, $C_{10}H_{17}NO_6$, and $C_7H_{10}O_4$) of primary products within the MCM, accounting for (on average) 69.0 ± 10.8% of the signal detected in the gas phase. Here some possibilities are plausible, one could be that they are produced as monomer but are important building blocks in the dimer formation, thus thermally decompose back to monomers during desorption."*

line s 367-369: not sure how / why this suggests favored dimer formation – confusing

**Action:** Se point above

line 370: another new designation, "high-MW"! dimer/oligomer? be consistent in how you refer to the different groups

**Action:** Changed to dimer. (oligomer was removed in next sentence)

**Action:** Text now refers to Figure 4

line 374: do you mean "the monomer signal at higher N2O5 to limonene ratio"? If so, say so. If not, clarify what you mean

**Action:** The sentence has been modified and now reads:

"At high ratios of $N_2O_5$ to limonene, the fraction of dimer species decreased, whereas the percentage of monomer species (fragments) with high Tmax increased (Fig. 4)."

line 375-376: "The average . . . Fig. 4" : remove this sentence, not clear & not helpful, and can cite Fig. 4 above instead.

**Action:** The sentence has been removed

line 407: define oxidation state here, where you first mention it, instead of later where you currently define it Fig. 5: put the labels for mass, desorption temperature, O/C and oxidation state in the same order (1) on the plot, (2) in the caption (and mention all 4!), and put the y axes in the same order, to make life easier on your readers. Also, you show S.D. on the plot (I guess? or, what are the error bars?) – mention this in caption.

**Action:** Oxidation state now defined where first mentioned. The figure has been changed and the reference to SD is now given in the caption.

line 421 do you mean "and are also represented as members of cluster 0"? i.e., the ions are members of both clusters? Clarify.

**Action:** The text now reads

*"This results from the fact that 87% and 69% of cluster 1 and 2 ions, respectively, have secondary thermogram peaks and Tmax values, and the ions represented as members of both clusters 1 and 0"*

*line s 424-428: Could an alternative rationale for this difference in oxidation state simply be that monomers, because of their small carbon chain length, need more oxidized functional groups to condense, while dimers are so big they'll condense even with less oxidation? Since you are looking only at the aerosol phase here, this pattern could be skewed by differing volatility.*

**Action:** Yes, this could be one reason. This alternative rationale is now included in the discussion.

*"It could be that monomers need more oxidation before being transferred into the condensed phase. However, as outlined by the partitioning plots (Fig. S1 most monomers also have a significant condensed phase contribution. Rather, this observation provides some insight into the processes of dimerization that are occurring, indicating the extent to which oxygen is lost during the dimerization process."*

*line 437: oxidation state is defined here, should be at first instance of it earlier in text.*

**Action:** Oxidation state now defined at first instant.

*line 438: @ "certain range of masses", state the range*

**Action:** The wording has been changed since there also was a mixup in Figure 5.

*"A positive trend between the Mw and Tmax values, see Fig 5a., was obtained for data in two of the monomer clusters (1 and 2) and the high volatile dimer cluster, while the trend turned negative for the low volatile dimers cluster. It should be noted that monomer species had (in general) higher O/C ratios than the dimers. It could be that monomers need more oxidation before being transferred into the condensed phase. However, as outlined by the partitioning plots (Fig. S1) most monomers also have a significant condensed phase contribution. Rather, this observation provides some insight into the processes of dimerization that are occurring, indicating the extent to which oxygen is lost during the dimerization process."*

*line 440: same comment as line s 424-428.*

**Action:** ok, see comment on line 424-428

*line 445 "during the process, and"*

**Action:** This has been extensively revised due to other comments

*line 446: as mentioned in general comments above, I think it would be good to include a structural diagram of the HNO3 leaving reaction.*

**Action:** Even if the mechanism on $HNO_3$ as the leaving group (in a reverse esterification process) is plausible we now after the review comments would be a little bit more cautious. This cautiousness has now been implemented in the manuscript and in line with that we do not want to be explicit in this formation. However, we include that this could be seen as a reverse esterification but also might be linked to observed hydrolysis of organic nitrates (Rindelaub et al, 2015, 2016).

*"The mechanism to create dimers with one nitrogen and a lower O/C ratio would presumably involve the loss of a nitrogen oxides or nitric acid. For this complex system and within the scope of this study it was not possible to firmly proof any mechanism. Since the experiment were done at low RH the direct hydrolysis would be less likely (see Rindelaub et al, 2015, 2016). However, knowing $HNO_3$ being thermodynamic stable one may speculate in that dimerization of two monomer species via the loss of one $HNO_3$ molecule could occur e.g. where a $C_{20}H_{29}NO_y$ (y = 10–18) species would be generated from*

*$C_{10}H_{15}NO_x$ (x = 5–9) species. This process could be seen as the reverse of esterification in order to produce a dimer product with one less nitrogen and reduced numbers of oxygens."*

line s 450 & 452: since this relied on having C9 monomers, it would be useful also to explain how these are formed in the mechanism leading up to dimerization.

**Reply:** There are several pathways for C9 monomer compounds (e.g. 11 MCM compounds has formula $C_9H_{14}O_4$). E.g. if both double bonds are oxidizes and there is a C-C bond breakage for the exocyclic double bond it will create a C9 monomer. Additional oxygens could be produced by internal H abstractions (e.g. isomerization of RO or $RO_2$).

line 453: at the end of this discussion, I think it would also be good to at least mention any alternative possible pathway that could make these dimer products – is it really only possible with HNO3 loss, or could you get it some other way? Then, perhaps the nitrogen balance or other evidence can help you bolster your hypothesis that the HNO3 loss is the more likely route.

**Reply/Action:** Yes, there are some suggestions in the literature, however, it is not as clear as one hope it should be and we cannot firmly conclude this so we have to leave that open for speculations and following up-studies on simpler systems. The nitrogen budget would be useful but since $HNO_3$ is notorious to be "sticky" and we did not measure the other suspected "leaving groups" like $NO_2$ and NO we have poor handled on this. However, it is clear that the selected dimers only contain one nitrogen and have lower O/C ratio than the sum of the two monomers. Furthermore, the Tmax suggests a high Mw compound (with low vapor pressure) and not a fragment. We now refer to some recent literature on organic nitrate condensed phase reactions (e.g. Rindelaub et al, 2015, 2016) as alternative way to loose nitrogen by hydrolysis.

*"The mechanism to create dimers with one nitrogen and a lower O/C ratio would presumably involve the loss of a nitrogen oxides or nitric acid. For this complex system and within the scope of this study it was not possible to firmly proof any mechanism. Since the experiment were done at low RH the direct hydrolysis would be less likely (see Rindelaub et al, 2015, 2016). However, knowing $HNO_3$ being thermodynamic stable one may speculate in that dimerization of two monomer species via the loss of one $HNO_3$ molecule could occur e.g. where a $C_{20}H_{29}NO_y$ (y = 10–18) species would be generated from $C_{10}H_{15}NO_x$ (x = 5–9) species. This process could be seen as the reverse of esterification in order to produce a dimer product with one less nitrogen and reduced numbers of oxygens."*

line s 463-464: "dimer fragmentation . . . monomers." is a confusing sentence

**Action:** Fragmentation has been replaced and the sentence reworded. The sentence now reads:

*"Due to the loss of $HNO_3$ during dimerization, the potential dimer decomposition during desorption is expected to yield fragments which differ in molecular composition from the precursor (i.e., pre-dimerization) monomers."*

line s 465-469: can't you demonstrate this more conclusively by looking at specific examples of masses with and without double peaks, and possibly matching up the Tmax's, and thus identify the subset of monomers that are also dissociation products and possibly connect them to precursors? Even better – if relative amounts change with different reaction conditions, can you track them rising and falling together? This dataset would seem to have lots of potential to demonstrate these actually linkages, not just make vague statements and what might / could yield what else upon fragmentation.

**Reply/Action:** In theory, this could be a way forward. However, the complexity and numbers of free parameters will put constraints on such detailed analysis. Actually, the suggestion is very nice but should be done on simpler systems where more solid conclusions on this mechanism could be done. The suggested mechanism is now brought forward with cautiousness and this is now supported with statements on other possible leaving groups and reference to previous work. Hopefully, it will inspire further work on this mechanism.

*"The mechanism to create dimers with one nitrogen and a lower O/C ratio would presumably involve the loss of a nitrogen oxides or nitric acid. For this complex system and within the scope of this study it was not possible to firmly proof any mechanism. Since the experiment were done at low RH the direct hydrolysis would be less likely (see Rindelaub et al, 2015, 2016). However, knowing $HNO_3$ being thermodynamic stable one may speculate in that dimerization of two monomer species via the loss of one $HNO_3$ molecule could occur e.g. where a $C_{20}H_{29}NO_y$ (y = 10–18) species would be generated from $C_{10}H_{15}NO_x$ (x = 5–9) species. This process could be seen as the reverse of esterification in order to produce a dimer product with one less nitrogen and reduced numbers of oxygens."*

line 480: "obtained for both gas and condensed-phase" – since you don't discuss gas phase data in this paper, omit, or cite to the companion paper that does study gas phase?

**Action:** Rephrased to state only particle phase

*"High-resolution mass spectrometric data was analysed for condensed-phase reaction products resulting from $NO_3$ initiated oxidation of the monoterpene, limonene."*

line 484-485: lots of sig figs on these percentages considering the error bars – better to say 63 +/- 7 and 37 +/- 7 %? And perhaps put a few numbers in the abstract?

**Action:** Number of significant figures has been reduced line 494-496: this has me wondering whether you can learn anything from the relative intensities of the two peaks? would this pattern support that SOA is "largely determined" by low-volatility oligomers?

**Reply:** If the two peaks are two different compounds with two different formation mechanisms and relative contribution to SOA there is little connection between them as for any pair of compounds detected. So we would not draw those conclusions from these observations.

line 504-505: as mentioned in the general comments, I think you should at least do some discussion of the non-listed products and what they might be.

**Action:** The products that were not found in the MCM list are now marked in the full list found in the supplemental. A comment on these and potential sources are now included.

*"There are two frequently suggested pathways for these. Firstly, the high number of oxygens would be result of isomerization of RO or $RO_2$ that rarely is described explicit in current modelling framework. Secondly, the presence of di-nitrated compounds relies on secondary chemistry derived from e.g. produced mononitrates intermediates; for limonene containing two double bonds this is more relevant than for other monoterpenes and so far not commonly described in models."*

line s 509-512: are all the hypothesized HNO3-loss dimers in one cluster and others in another? Or, how else can you use the cluster analysis to learn something related to your mechanism speculation?

**Reply:** The cluster analysis clearly separated the fragments from the dimers (obviously, since a fragment cannot be formed according to the mechanism) and also classified the dimers in two groups. However, ions from dimer families containing two nitrogen's were found in both clusters. (see Table 2).

**Action**: The explicit statement on the direct link between cluster analysis and the mechanism has been removed.

line 513: "FIGAERO" to be consistent with how you write it above

**Action:** Text changed to read "FIGAERO-CIMS"

line 514: "may provide some means of reducing the complexity. . ." see above general comments. I hope you can work get a bit more out of this.

**Reply:** The changes in the manuscript induced by reviewers have certainly improved the quality of the manuscript and the analysis. Part of this conclusion is hope for some further work to enable a more stringent description on SOA induced from $NO_3$ chemistry.

line 520-521: as mentioned above, hard to include species in modeling studiest if you haven't even suggested what you think they might be.

**Action:** The compounds are now listed in the supplemental and a short note on potential for extension of model descriptions is given.

*"There are two frequently suggested pathways for these. Firstly, the high number of oxygens would be result of isomerization of RO or $RO_2$ that rarely is described explicit in current modelling framework. Secondly, the presence of di-nitrated compounds relies on secondary chemistry derived from e.g. produced mononitrates intermediates; for limonene containing two double bonds this is more relevant than for other monoterpenes and so far not commonly described in models."*

**Response to Anonymous Referee #3**

This work investigated organic nitrate formation from NO3 oxidation of limonene. Experiments were conducted using different N2O5/limonene ratios. Speciated gas and particle phase organic nitrates were measured by the FIGAERO-HR-ToF-CIMS. Cluster analysis of the desorption temperatures of organic nitrate species resulted in five clusters; the relationships between O/C, OS, MW, etc of these clusters were discussed. Formation of dimers was observed and reaction mechanisms for dimer formation were proposed.

This is an interesting study and the manuscript is generally well-written and easy to follow. This study will be of interest to the greater atmospheric community. My main comments are 1) while the experiments were conducted over a range of N2O5/limonene ratios, the authors shall provide more context to this experimental design. Also, the results from experiments with different N2O5/limonene ratios need to be more extensively and clearly discussed. 2) I have some concerns regarding the discussion of the results shown in Fig. 5, please see details below. 3) There are a number of recent studies on nitrate radical oxidation of biogenic hydrocarbons, it would be appropriate that these studies are referenced in the manuscript to reflect the current state of knowledge. Overall, I recommend publication in ACP once these comments are addressed. Most comments are mainly to improve clarity of the manuscript.

**Main Comments**

1. Page 5 line 158. Are potential impurities (e.g., NO2 and HNO3) in the N2O5 synthesized measured and quantified? Please make this clear in the manuscript.

**Action:** This is now clarified in the experimental:

"$N_2O_5$ was synthesized by reacting ≥20 ppm $O_3$ with pure $NO_2$ (98%, AGA Gas) in a glass vessel and then passing the flow through a cold trap maintained at -78.5 ℃ using dry ice. Even if neither $HNO_3$ nor $NO_2$ was measured it is known from previous work that this method typically provides a source with impurities less than a few percent. It is well known that the resulting white solid would show signs of yellowing, due to nitric or nitrous acid contamination, if exposed to moisture (e.g., ambient lab air) so handling of the $N_2O_5$ was done accordingly."

2. Page line 168. What is the reason for performing experiments with different N2O5/limonene ratios? Please provide more context here.

**Reply:** It would reflect a gradual increase in reaction of the exocyclic double bond. At a ratio of 1:1 we expect $NO_3$ to primarily react with the exocyclic double bond, representing the typical case producing primary products, while increasing the ratio will add more possibilities for reactions also with the endocyclic double bond, reflecting secondary atmospheric chemistry.

**Action:** Added text in experimental:

"At a ratio around 1.0 one expects only the endocyclic double bond to be reacting with $NO_3$ radicals while at higher ratio there is an increased possibility for secondary chemistry where products will be susceptible for reaction with the $NO_3$ radical."

3. Page 6 Table 1. a. With these N2O5/limonene ratios, are all limonene (and both double bonds?) reacted away? Please clarify and change the "limonene" in the table to either "reacted limonene" or "initial limonene". b. What is the RO2 reaction regime in these experiments? RO2 + NO3? RO2 + RO2? c. Can SOA yields be quantitatively calculated from the values in the table? If the "limonene" is reacted limonene, the SOA yields appear to be very low compared to previous studies by Fry et al. (ACP, 2009), Fry et al. (ACP, 2011), and Boyd et al. (ES&T, 2017). Please discuss the results from this study in the context of these prior studies. Also, do the data shown in Table 1 follow a typical Odum 2- product yield curve?

**Reply:** If not necessary we prefer not to explicitly present aerosol yields from this flow reactor study. The experiments were not designed to derive yields.

4. Page 8 line 238. It was noted that ". . ..the relative signal intensities varied with the amount of limonene and N2O5 present in the system". I think the authors are referring to Fig. 4? Please add the figure number to the sentence to help guide the readers if this is the case.

**Reply:** Yes, the referee is right but we then realize that this figure would then be introduced too early in the manuscript.

**Action:** We have removed the statement

"and the relative signal intensities varied with the amount of limonene and $N_2O_5$ present in the system."

5. Page 8 line 260. It is not immediately clear what these species are without diving into the entire MCM mechanisms. The authors should at least include the formation mechanisms of these major ions in the SI to help guide the readers. Also, it would be helpful to propose mechanisms for the major species that were observed in this study but are not in MCM. On the related note, Boyd et al. (ES&T, 2017) recently expanded the limonene + NO3 mechanisms in MCM. It might be worthwhile to evaluate if some species detected in this study are covered in the expanded mechanism in Boyd et al.

**Reply:** The Boyd et al. paper was unfortunately not available when the majority of the analysis in the present study was done.

**Action:** We have referred to the Boyd et al, study. Even if we here and there refer to MCM compounds we have made it easier by listing MCM compounds in the full list of the 198 ions identified (supplemental). Furthermore, in addition to the described dimer mechanism speculation we summaries the general pathway for mechanism developments by including the following statements:

"The non-listed species (see Table S1) were either dimer species or more highly oxygenated, nitrated analogs of known major products, which are notoriously hard to describe via standard gas-phase mechanisms. There are two frequently suggested pathways for these. Firstly, the high number of oxygens would be result of isomerization of RO or $RO_2$ that rarely is described explicit in current modelling framework. Secondly, the presence of di-nitrated compounds relies on secondary chemistry derived from e.g. produced mononitrates intermediates; for limonene containing two double bonds this is more relevant than for other monoterpenes and so far not commonly described in models."

6. Page 9  line  280. What are some of the mechanisms for limonene and its oxidation products to react with NO2 and HNO3?

**Reply:** The possibilities are for example that $NO_2$ could react with peroxy-radicals and $HNO_3$ might do condensed phase nitration. We don't think that this of general value and would not add that.

7. Page 10  line  293. Nah et al. (ES&T 2016) also measured a large suite of highly oxygenated organic nitrates from NO3 oxidation of a-pinene and b-pinene in laboratory experiments, using the FIGAERO-HR-ToF-CIMS. Many of those are also observed in Lee et al. (PNAS, 2016).

**Reply:** We have noted the request for including Nah et al. (ES&T 2016).

**Action:** As the last sentence in this paragraph we added:

*"Complementary, Nah et al. (2016) also measured a large suite of highly oxygenated organic nitrates from $NO_3$ oxidation of a-pinene and b-pinene in laboratory experiments."*

8. Page 10  line  299. It is noted that "The similarity with ions from the NO3-initiated limonene oxidation further emphasizes the importance of monoterpenes as precursors of organic nitrates". It would be informative if the authors indicate in the Table in SI (the ion list) regarding which ions have also been observed in the ambient (Lee et al.) and other monoterpene experiments (Nah et al).

**Reply:** We would prefer to focus on comparison with limonene studies. It turns out there is now a parallel study by Boyl et al, 2017 (accepted when this manuscript was submitted) that could be better suited for comparisons.

9. Page 10 line  301. Rollins et al. (Science, 2012) discussed the importance of limonene + NO3 in a field study.

**Action:** This reference has now been added in the instruction. This part has been reduced, as a consequent of other comments, and do not refer to the importance of limonene+$NO_3$

10. Page 12  line  362. It was noted that there is a positive correlation (R2 = 0.67) between Tmax and molecular mass. Are the authors referring to Fig 5a? If so, it does not look like the overall correlation is that good? Please clarify.

**Reply/action:** Yes, it referred to Fig5a. However, its better to have that discussion later in the paper so we removed the statement here and extended/clarified the text later to read.

*"A positive trend between the Mw and Tmax values, see Fig 5a., was obtained for data in two of the monomer clusters (1 and 2) and the high volatile dimer cluster, while the trend turned negative for the low volatile dimers cluster."*

11. Page 13  line  370-379. (this is also related to comment #2 above). It is not clear to me how the higher N2O5/limonene experiments lead to formation of more thermally unstable mechanisms. Please explain. More discussions are needed here, to provide context to why experiments are conducted with different N2O5/limonene in the first place and why/how the resulting compositions are different.

**Reply:** This is discussed in the above #2 comment

12. Page 13 Figure 4. What is the function used for the fit? Is this just to guide the eye or there is a fundamental reason for such a dependence?

**Reply:** yes, we agree the line is mostly for guidance of the eye and the r2 should be removed.

**Action:** The r2 has been removed. The end of the Figure caption now reads:

*"The lines indicated are for the guidance of the eye."*

13. Page 15, discussions of Figure 5. The authors attempted to discuss the relationships between O/C, OS, and MW, etc. However, within uncertainties, there do not seem to be significant differences in the O/C and OS values for all clusters. Hence, this discussion needs to be revised.

a. (related to comments # 2 and 11 above). In Figure 5, will any specific patterns emerge if the authors only look at the data from experiment of a particular N2O5/limonene ratio?

b. line 418. It was noted that the O/C of cluster 0 is similar to clusters 1 and 2. However, within the uncertainties, the O/C ratios of all clusters are almost the same.

c. line 426, should the high-MW clusters be (3,4)? And the low-MW clusters be (0, 1, 2)?

d. line 427. It was noted that the ions in the high-MW clusters have a lower OS than ions in clusters 0-3. Firstly, should "0-3" be "0-2"? Secondly, it does not look like the high-MW clusters have a lower OS. Within uncertainties, the OS values appear to be the same for all clusters.

e. line 434. It was noted that a positive correlation exists between O/C and Tmax. It is not clear how this is case from the data shown in Fig. 5c. Please provide a figure and show the R2 value.

**Action/replies:**

   a) Figure 5 describes a cluster analysis that requires multiple data and cannot be compared with single experiments. Generally, the same features are found for individual experiments. However, there is a gradual change when changing the N2O5/limonene as illustrated in Fig 4.
   b) The SD is not necessary described as an uncertainly but rather the variability in each property. It's a clear and significant shift in the median for each property even if the clusters have overlapping variability.
   c) Yes, text changed accordingly
   d) Yes, typo changed. For uncertainty discussion see b)
   e) The figure has been mixed up. Intention was to describe the trends in Fig 5a, i.e. Mw vs Tmax. The text has been changed and now reads:
      *"A positive trend between the Mw and Tmax values, see Fig 5a., was obtained for data in two of the monomer clusters (1 and 2) and the high volatile dimer cluster, while the trend turned negative for the low volatile dimers cluster."*

**Minor Comments**

1. Page 2  line  44. Would be appropriate to reference Ng et al. (ACP, 2017).

**Action:** Reference has been added

2. Page 2  line  46. Would be appropriate to also reference Day et al. (AE, 2010); Fry et al. (ACP, 2013); Xu et al. (PNAS, 2015); Xu et al. (ACP, 2015); Boyd et al. (ACP, 2015); Kiendler-Scharr et al. (GRL, 2016); Nah et al. (ES&T, 2016).

**Action:** References have been added

3. Page 2  line  60. Delete "M" in front of Hallquist.

**Action:** The "M." has been removed

4. Page 3  line  77. Boyd et al. (ES&T, 2017) recently investigated SOA formation from NO3 oxidation of limonene.

**Action:** Reference has been added

6. Page 14 Figure 5. Missing y-axis label for 5b?

**Action:** This is fixed now.

[revised manuscript text omitted]

**Page 6: [1] Deleted**                                             **Author**

| | | | | |
|---|---|---|---|---|
| 1 | 160 | 15 | 10.7 | 8.1 |
| 2 | 95 | 40 | 2.4 | 8 |

**Page 6: [2] Deleted**                                             **Author**

| | | | | |
|---|---|---|---|---|
| 11 | 850 | 95 | 8.9 | 25 |
| 12 | 850 | 150 | 5.7 | 47 |